psychology

smartphone use, phenomenology, motivation, opportunity costs, work

**Author for correspondence:**
Jonas Dora
e-mail: jonas.dora.psych@gmail.com

# Fatigue, boredom and objectively measured smartphone use at work

Jonas Dora, Madelon van Hooff, Sabine Geurts, Michiel Kompier and Erik Bijleveld

Behavioural Science Institute, Radboud University, Nijmegen, The Netherlands

JD, 0000-0003-0651-8479; EB, 0000-0001-5892-3236

Nowadays, many people take short breaks with their smartphone at work. The decision whether to continue working or to take a smartphone break is a so-called labour versus leisure decision. Motivational models predict that people are more likely to switch from labour (work) to leisure (smartphone) the more fatigue or boredom they experience. In turn, fatigue and boredom are expected to decrease after the smartphone was used. However, it is not yet clear how smartphone use at work relates to fatigue and boredom. In this study, we tested these relationships in both directions. Participants ($n = 83$, all PhD candidates) reported their current level of fatigue and boredom every hour at work while an application continuously logged their smartphone use. Results indicate that participants were more likely to interact with their smartphone the more fatigued or bored they were, but that they did not use it for longer when more fatigued or bored. Surprisingly, participants reported increased fatigue and boredom after having used the smartphone (more). While future research is necessary, our results (i) provide real-life evidence for the notion that fatigue and boredom are temporally associated with task disengagement, and (ii) suggest that taking a short break with the smartphone may have phenomenological costs.

## 1. Introduction

Nowadays, most people own a smartphone [1] and have it within reach throughout the day. Just two decades ago, people were mostly disconnected from their private life while at work; today, people can use their smartphone for private matters during working hours [2]. Indeed, even when they are at work, smartphones enable people to be constantly connected to friends and family through social media and instant messaging applications [3]. Against the background of this societal development, here we study the relationship between aversive subjective experiences that are indicative of low

motivation—i.e. fatigue and boredom—and smartphone behaviour at work. We study this relationship in both directions. We test whether fatigue and boredom predict greater subsequent smartphone use at work; also, we test whether fatigue and boredom decrease when people have just used their smartphone.

Both for science and for practice, it is potentially important to examine the relationship between (i) fatigue and boredom and (ii) smartphone use. For science, this study provides a real-life test of motivational theories of fatigue and boredom, which so far have been mostly studied in the laboratory. For practice, this study may guide the development of interventions aimed at maximizing the potential benefits of smartphone use at work (e.g. improved motivation and recovery).

It is currently not yet clear how fatigue and boredom relate to smartphone use during worktime. On the one hand, previous research has uncovered some negative effects of smartphones for task engagement. That is, in both field [4] and laboratory [5,6] studies, smartphone notifications were found to harm people's ability to concentrate on their current task. It seems plausible that fatigue and boredom augment these negative effects. Indeed, in laboratory studies, fatigued participants were more likely to disengage from their main task in order to engage with their smartphone [7]. On the other hand, smartphone interactions may serve a more positive function, in that they may act as *microbreaks*, during which people can recover from work demands [8,9]. Consistent with this idea, laboratory studies suggest that smartphone interactions have recovery potential, especially for people who enjoy interacting with their smartphone [7]. So, although laboratory research has produced useful insights, it remains an open question whether fatigue and boredom are related to smartphone behaviour in real life.

Our hypotheses are grounded in recent motivational models of fatigue and boredom. We assume that to continue working versus to use one's smartphone represents a goal conflict—specifically, a conflict between *labour goals* and *leisure goals* [10,11]. In the context of cognitive work, a labour task is any activity that is productive but mentally demanding (e.g. grading a thesis); a leisure task is any activity that is unproductive and mentally undemanding (e.g. answering a friend's text message). According to motivational models of fatigue and boredom, the shared function of these experiences is to resolve such goal conflicts [10,12–15]. Specifically, fatigue and boredom should arise when the current (labour) task is judged to have lower value than some alternative (leisure) task. In other words, conscious feelings of fatigue and boredom are thought to reflect a discrepancy between what is currently being done and what should be done instead. Both fatigue and boredom are defined as aversive subjective experiences [13,16]. The difference between fatigue and boredom is assumed to depend on the amount of stimulation currently provided by the (labour) task. When people invest a lot of cognitive resources into the current task while this task is judged to be less valuable than some alternative, people should experience fatigue [14]; when the current task provides insufficient cognitive stimulation and the current task is judged to be less valuable than some alternative, people should experience boredom [15].

Previous research has provided preliminary evidence for a role of motivation in the experiences of fatigue and boredom of workers. For example, two diary studies [17,18] showed that nurses and university staff, respectively, experienced lower levels of fatigue the more they report to find their work rewarding and pleasurable. Additionally, two small studies [19,20] showed that the experience of boredom can be significantly reduced through the short interaction with an enjoyable alternative task (such as the smartphone).

In line with this idea, we hypothesize that (more) fatigue and boredom is related to (more) subsequent smartphone use at work (hypothesis 1). Engaging in leisure for some time should update the balance between labour and leisure [10]. Specifically, after engaging in leisure (e.g. using one's smartphone), the relative value of labour (e.g. working on a spreadsheet) should increase, which should decrease fatigue and boredom [14]. In line with this rationale, we hypothesize that after more smartphone use, people experience less fatigue and boredom (hypothesis 2).

To test our hypotheses, we conducted an experience-sampling study in which participants (PhD candidates who owned an Android smartphone, reported high job autonomy and reported to use their smartphone more for private matters than for work-related matters during working hours) rated their current level of fatigue and boredom, every full hour while they were at work, for three working days. At the same time, an application on participants' smartphone continuously monitored their smartphone use.

The use of momentary measures of fatigue and boredom as well as a monitoring application to quantify smartphone use allows us to improve on previous research indicating correlations between subjective experiences and smartphone use [21,22]. These studies, which suggested that participants who generally experience more boredom also use their smartphone more, relied on cross-sectional

assessments of subjective experiences and self-reported smartphone use. Previous research [23,24] has shown that self-reported smartphone use does not correlate meaningfully with actual smartphone use. Thus, by linking state-level self-report data on fatigue and boredom with momentary objective smartphone use data, we were able to model the moment-to-moment temporal relationship of these affective states with objective smartphone use within participants, and vice versa.

Our predictions rest on the assumption that the smartphone is a highly valued leisure alternative to labour [14]. More specifically, this opportunity cost model predicts that the relationship between the subjective experiences of fatigue and boredom depends on the value of the leisure alternative to work (e.g. the smartphone). Thus, our predicted effects should be stronger for participants who value their smartphone (interactions) more. Research indicates that people differ in their desire to stay continuously connected to their friends and family via the Internet [25]. This individual difference, labelled *fear of missing out* (FOMO), is thought to reflect the degree to which people value to stay connected to others through digital technology [25]. Hence, higher FOMO should be associated with a higher value of the current leisure alternative. For that reason, we additionally tested whether individual differences in FOMO strengthen the relationships of smartphone use with fatigue and boredom in both directions.

# 2. Results

Participants ($n = 83$) responded to 1724 hourly self-reports of fatigue and boredom (20.77 per participant on average). We calculated smartphone use 20 min after and 20 min before these self-reports to examine the effect of fatigue and boredom on subsequent smartphone use, and vice versa. After excluding data points in which this 20 min timeframe pre- or post-questionnaire overlapped with the start time of work, end time of work or lunch break, we were left with 1477 (pre) and 1482 (post) data points, respectively.

## 2.1. Descriptive statistics

In the 20 min following the completion of the hourly questionnaire, participants had interacted with their smartphone in 52% of the cases. On average, they spent 92 s (s.d. = 181) on their smartphone, which equals approximately 7% of the timeframe ($M = 84.03$, s.d. = 168.16 in the 20 min preceding the completion of the hourly questionnaire). The distribution of smartphone use in the 20 min post-questionnaire interval is shown in figure 1*a*. Smartphone use is non-normally distributed, mainly owing to a higher number of time intervals during which participants did not interact with their smartphone at all. The development of smartphone use over the working day is shown in figure 1*b*. Across all hourly questionnaires from all participants, the mean fatigue was 27 points (on a 100-point visual analogue scale; s.d. = 22). The development of fatigue over the working day is shown in figure 1*c*. Across all hourly questionnaires from all participants, the mean boredom was 17 points (on a 100-point visual analogue scale; s.d. = 19). The development of boredom over the working day is shown in figure 1*d*. Overall, both fatigue and boredom increased over the course of the working day. This increase in fatigue is not quite linear owing to participants reporting slightly elevated levels of fatigue in the early morning hours. The same trend is not observed in boredom, which steadily increased throughout the day. The correlation between fatigue and boredom on an hourly level was 0.45 and is shown in figure 1*e*. Participants on average reported medium levels of FOMO (on a 5-point Likert scale; $M = 2.63$, s.d. = 0.98). The distribution of FOMO in our sample is shown in figure 1*f*.

## 2.2. Effect of fatigue and boredom on subsequent smartphone use

To test the effect of fatigue and boredom on subsequent smartphone use, we fitted two Bayesian mixed-effects models. With a binomial model, we first tested whether greater fatigue and boredom make it more likely that the smartphone is *used at all* in the following 20 min. Next, with a zero-inflated $\beta$ model, we tested whether greater fatigue and boredom are related to more smartphone use (i.e. time spent interacting with the smartphone) in the following 20 min *in case the smartphone is used at all* (i.e. if smartphone use is greater than 0).

Figure 2*a* shows the posterior distributions of our model of the effect of fatigue on the subsequent likelihood that the smartphone was used, overlaid with the posterior mean and 95% Bayesian credible interval. For this plot, we transformed the parameters of the model to the odds ratio. The posteriors

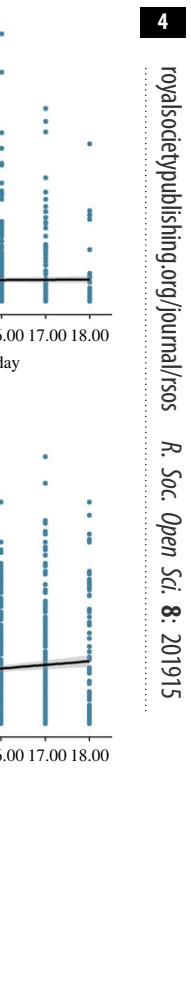

**Figure 1.** Descriptive statistics. The dotted line in (*a*) represents the average smartphone use during 20 min at work.

show that as fatigue increases by half a standard deviation (approx. 11 points), participants are estimated to be 1.36 times more likely to interact with their smartphone in the following 20 min (95% credible interval = (1.02, 1.82)). Neither FOMO nor the interaction between fatigue and FOMO seemed to have any notable effect on the likelihood to use the smartphone, with the mean of the posterior being close to 1. Figure 2*b* shows the raw data associated with the effect of fatigue on the probability to use the smartphone.

Similarly, figure 3*a* shows the posterior distributions of our model of the effect of boredom on the subsequent likelihood that the smartphone was used. The posteriors show that as boredom increases by half a standard deviation (approx. 10 points), participants are estimated to be 1.43 times more likely to interact with their smartphone in the following 20 min (95% credible interval = (1.09, 1.92)). Again, neither FOMO nor the interaction showed any effect on the subsequent likelihood to interact with the smartphone. We conclude that there is some evidence for a small effect of fatigue and boredom on the decision to interact with the smartphone, which is consistent with hypothesis 1. Figure 3*b* shows the raw data associated with the effect of boredom on the probability to use the smartphone.

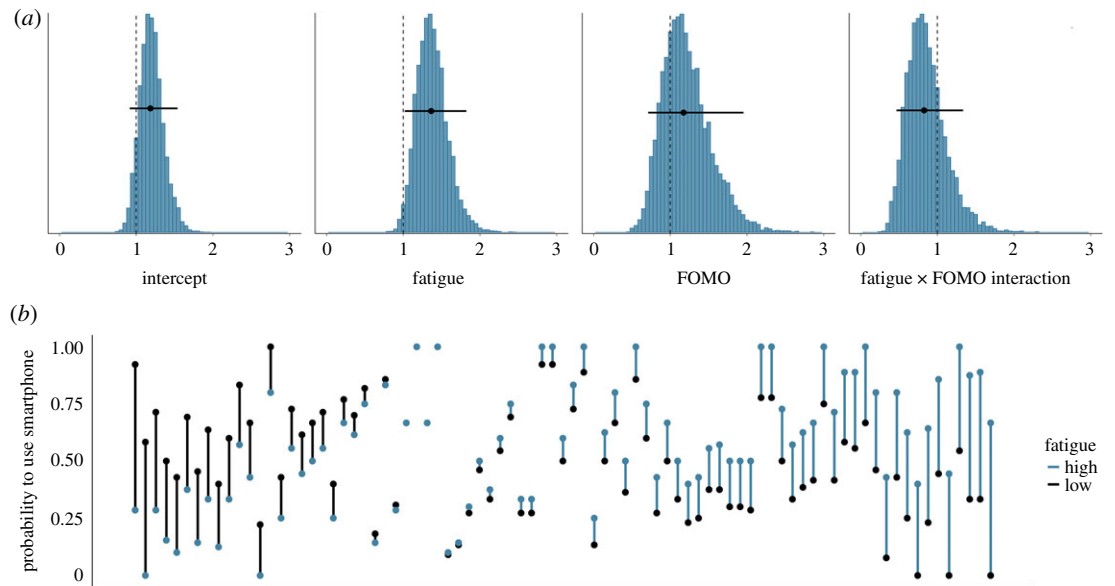

**Figure 2.** (*a*) The exponentiated posterior distributions of the parameters (reflecting the odds ratios) for the fatigue model predicting subsequent likelihood to use the smartphone. The circles and the lines represent the mean of the posterior and the 95% Bayesian credible intervals, respectively. (*b*) The probability of smartphone use in the 20 min timeframe, separately for each participant. Each participant is represented by one line. Black dots indicate participants' probability of smartphone use when they are relatively low in fatigue (i.e. below their own mean level); blue dots represent participants' probability of smartphone use when they are relatively high in fatigue (i.e. above their own mean level). Black (blue) lines indicate that participants were more likely to use the smartphone when fatigue was low (high). Participants are arranged by the magnitude of the relationship of fatigue on subsequent likelihood to use the smartphone, from left (higher probability to use smartphone when fatigue is lower) to right (higher probability to use smartphone when fatigue is higher).

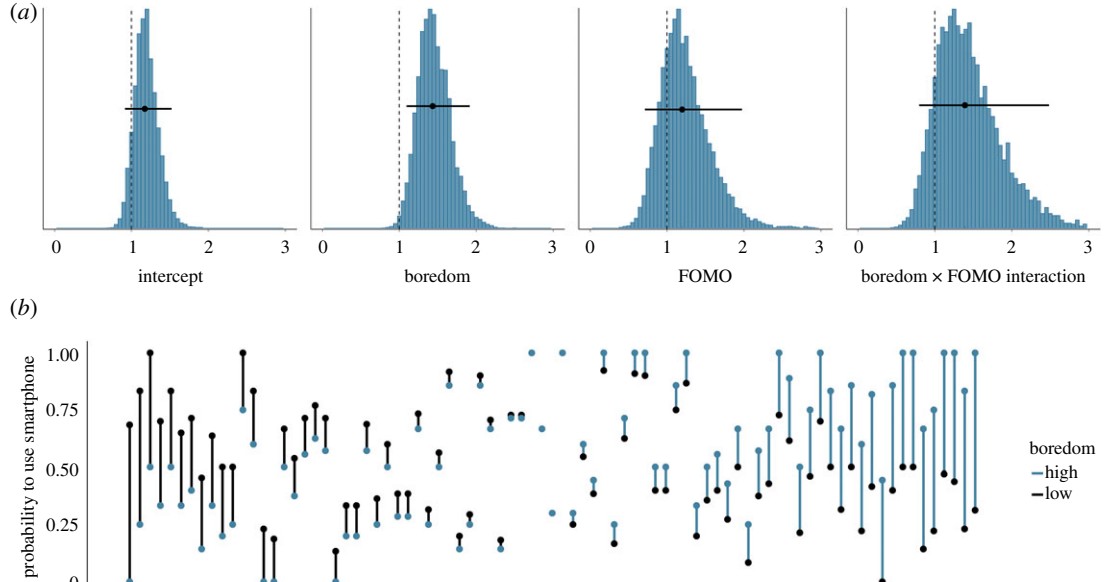

**Figure 3.** (*a*) The exponentiated posterior distributions of the parameters (reflecting the odds ratios) for the boredom model predicting subsequent likelihood to use the smartphone. The circles and the lines represent the mean of the posterior and the 95% Bayesian credible intervals, respectively. (*b*) The probability of smartphone use in the 20 min timeframe, separately for each participant. Each participant is represented by one line. Black dots indicate participants' probability of smartphone use when they are relatively low in boredom (i.e. below their own mean level); blue dots represent participants' probability of smartphone use when they are relatively high in boredom (i.e. above their own mean level). Black (blue) lines indicate that participants were more likely to use the smartphone when boredom was low (high). Participants are arranged by the magnitude of the relationship of boredom on subsequent likelihood to use the smartphone, from left (higher probability to use smartphone when boredom is lower) to right (higher probability to use smartphone when boredom is higher).

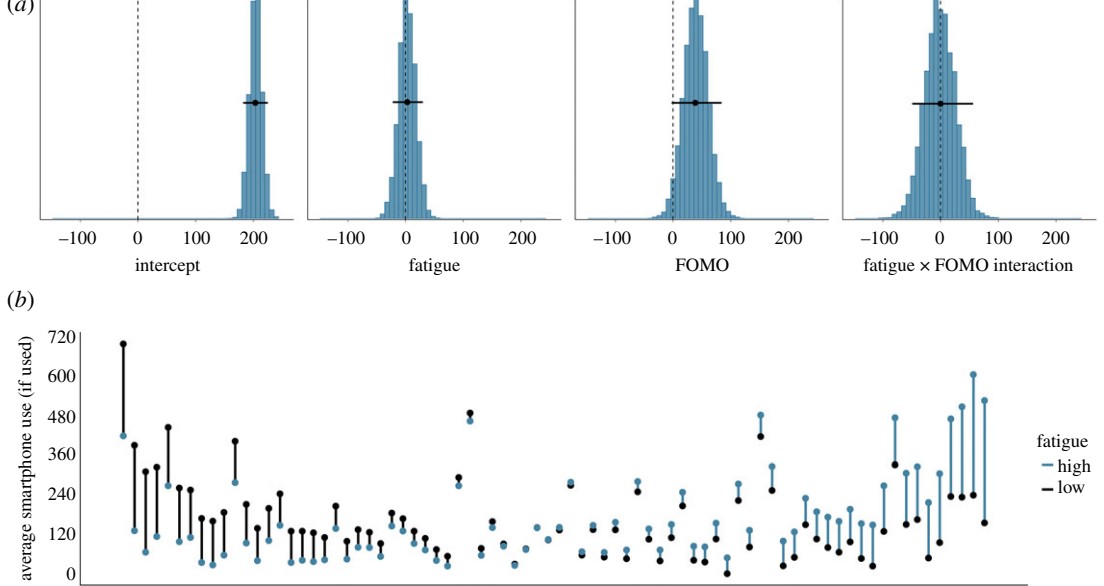

**Figure 4.** (a) The posterior distributions of the parameters for the fatigue model predicting subsequent smartphone use (in seconds) if the smartphone is used at all. The circles and the lines represent the mean of the posterior and the 95% Bayesian credible intervals, respectively. (b) The average smartphone use (in seconds) if the smartphone was used at all in the 20 min timeframe, separately for each participant. Each participant is represented by one line. Black dots indicate participants' average smartphone use when they are relatively low in fatigue (i.e. below their own mean level); blue dots represent participants' average smartphone use when they are relatively high in fatigue (i.e. above their own mean level). Black (blue) lines indicate that participants were using their smartphone more when fatigue was low (high). Participants are arranged by the magnitude of the relationship of fatigue on subsequent smartphone use, from left (higher smartphone use when fatigue is lower) to right (higher smartphone use when fatigue is higher).

Figure 4a shows the posterior distributions of our model of the effect of fatigue on smartphone use if the smartphone is used at all. The posteriors show that as fatigue increases by half a standard deviation, participants are estimated to use their smartphone for 2.8 s more in the subsequent 20 min (95% credible interval = (−22.22, 29.94)). The model further estimates that participants half a standard deviation higher than the average in FOMO use their smartphone for 39 s more in 20 min (95% credible interval = (−2.45, 84.24)). The mean of the posterior of the interaction between fatigue and FOMO is estimated to be close to zero. Figure 4b shows the raw data associated with the effect of fatigue on subsequent smartphone use if used at all.

Similarly, figure 5a shows the posterior distributions of our model of the effect of boredom on smartphone use if the smartphone is used at all. The posteriors show that as boredom increases by half a standard deviation, participants are estimated to use their smartphone for 10 s more (95% credible interval = (−15.90, 36.72)). Unsurprisingly, the model once more estimates that participants half a standard deviation higher than the average in FOMO use their smartphone for 37 s more in 20 min (95% credible interval = (−3.38, 81.51)). The model also estimates weak evidence that the relationship between boredom and subsequent smartphone use may be stronger for those participants lower in FOMO (mean$_{int}$ = −33.28, 95% credible interval = (−75.42, 16.94)). We conclude that there is no evidence for an effect of fatigue and boredom on the amount of time the smartphone is used, provided that it is used at all. This finding does not support hypothesis 1. However, there is some evidence that participants higher in FOMO use their smartphone more while at work. Figure 5b shows the raw data associated with the effect of boredom on subsequent smartphone use if used at all.

In the analyses presented above, we ran separate models predicting the smartphone use variables from fatigue and boredom. We used separate models for two reasons: first, on a theoretical level, fatigue and boredom should have the same functionality (i.e. both should motivate people to switch activities [14]), and we did not want to test the same mechanism twice in the same model. Second, on a methodological level, running separate analyses reduced the risk of overfitting our data [26]. That is, in our models, all random effects (i.e. individual per-participant and per-day intercepts and slopes) are introduced as additional parameters [27]. Moreover, as FOMO is considered to be a stable personality trait and was only assessed once, we had only one observation per participant for this

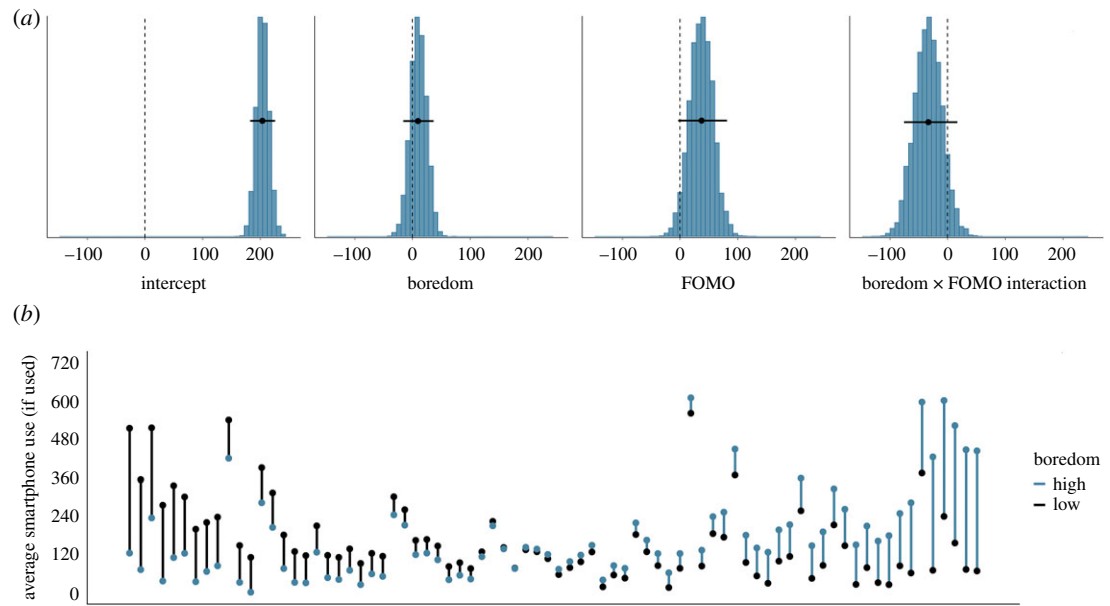

**Figure 5.** (*a*) The posterior distributions of the parameters for the boredom model predicting subsequent smartphone use (in seconds) if the smartphone is used at all. The circles and the lines represent the mean of the posterior and the 95% Bayesian credible intervals, respectively. (*b*) The average smartphone use (in seconds) if the smartphone was used at all in the 20 min timeframe, separately for each participant. Each participant is represented by one line. Black dots indicate participants' average smartphone use when they are relatively low in boredom (i.e. below their own mean level); blue dots represent participants' average smartphone use when they are relatively high in boredom (i.e. above their own mean level). Black (blue) lines indicate that participants were using their smartphone more when boredom was low (high). Participants are arranged by the magnitude of the relationship of boredom on subsequent smartphone use, from left (higher smartphone use when boredom is lower) to right (higher smartphone use when boredom is higher).

variable and the associated cross-level interactions. Thus, as including fatigue and boredom in one model would vastly increase the number of parameters, increasing the risk for overfitting, we chose against this strategy for our main analysis.

However, one weakness of the latter strategy is that it does not consider the potential synergy between fatigue and boredom in triggering smartphone use. Moreover, as the correlation between fatigue and boredom (on an hourly level) was only moderate (figure 1*e*), one could argue that fatigue and boredom probably do not reflect one single function. Thus, to explore this possibility that fatigue and boredom interact to predict smartphone use in a synergistic way, despite the risk of overfitting, we decided *post hoc* to fit two full models (predicting the subsequent likelihood that the smartphone is used and the amount of smartphone use) that included fatigue, boredom, FOMO, as well as all two-way and three-way interactions as independent variables. Results from these models are reported in detail in the electronic supplementary material. We briefly summarize the main findings here. First, parameter estimates from these more extensive models were comparable to those of the models reported in detail above. Like before, as fatigue and boredom increased by half a standard deviation, participants were estimated to be 1.28 and 1.40 times more likely to interact with their own smartphone in the subsequent 20 min, respectively. Second, fatigue and boredom did not predict the duration of those smartphone interactions. In neither model, we found evidence for any synergy between fatigue and boredom (e.g. it was not the case that boredom predicted the likelihood of smartphone use, especially when people were fatigued). Hence, whether we modelled the effects of fatigue and boredom separately or simultaneously, we arrived at the same interpretation of our data. We conclude from this additional analysis that fatigue and boredom have separate effects on the decision to disengage from work in order to interact with one's smartphone, and that these effects are not synergistic.

## 2.3. Effect of smartphone use on subsequent fatigue and boredom

To test the effect of smartphone use on subsequent fatigue and boredom, we again fitted two Bayesian mixed-effects models. First, we tested the effect of whether or not the smartphone was used at all in the 20 min prior to the hourly questionnaire on subsequent fatigue and boredom. Second, we tested the effect

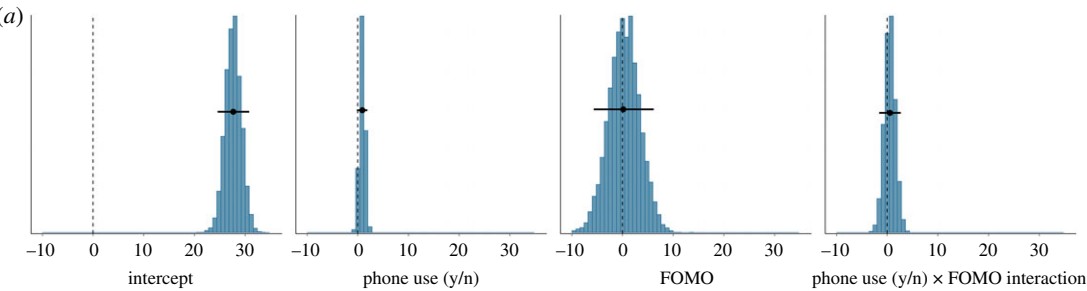

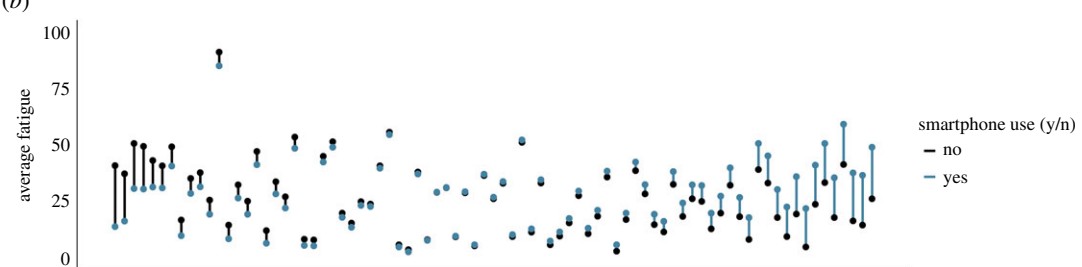

**Figure 6.** (*a*) The posterior distributions of the parameters for the smartphone use (y/n) model predicting subsequent fatigue. The circles and the lines represent the mean of the posterior and the 95% Bayesian credible intervals, respectively. (*b*) The average fatigue level, separately for each participant. Each participant is represented by one line. Black dots indicate participants' average fatigue when they did not use their smartphone; blue dots represent participants' average fatigue when they did use their smartphone. Black (blue) lines indicate that participants were more fatigued when they did not (did) use their smartphone. Participants are arranged by the magnitude of the relationship of smartphone use on subsequent fatigue, from left (higher fatigue when smartphone was not used) to right (higher fatigue when smartphone was used).

of total smartphone use in the 20 min prior to the hourly questionnaire on fatigue and boredom. Like before, we also tested whether these effects were strengthened by individual differences in FOMO.

Figure 6*a* shows the posterior distributions of our model of the effect of whether or not the smartphone was used on fatigue. The posteriors show that if the smartphone was used in the 20 min before the hourly questionnaire, fatigue is estimated to be higher by 0.89 points (95% credible interval = (−0.14, 1.91)). The effects of FOMO on fatigue and the interaction between whether or not the smartphone was used and FOMO are estimated to be close to zero. These results (as well as the results from the subsequent three models) did not change meaningfully when we controlled for time of day or fatigue at the previous hour, showing that the effect is not owing to general increases in fatigue and boredom over the course of the day or in the previous hour.[1] Figure 6*b* shows the raw data associated with the effect of whether or not the smartphone was used on subsequent fatigue.

Similarly, figure 7*a* shows the posterior distributions of our model of the effect of whether or not the smartphone was used on boredom. The posteriors show that if the smartphone was used, boredom is estimated to be higher by 1.32 points (95% credible interval = (0.39, 2.23)). Again, the effects of FOMO on boredom and the interaction between whether or not the smartphone was used and FOMO are estimated to be close to zero. We conclude that there is some evidence for a small effect of whether or not the smartphone was used on the subjective experiences of fatigue and boredom. However, this effect is in the opposite direction to hypothesis 2: smartphone use was associated with higher, not lower, subsequent fatigue and boredom. Figure 7*b* shows the raw data associated with the effect of whether or not the smartphone was used on subsequent boredom.

Figure 8*a* shows the posterior distributions of our model of the effect of total smartphone use on subsequent fatigue. The posteriors show that if smartphone use increases by half a standard deviation (approx. 80 s), fatigue is estimated to be higher by 0.88 points (95% credible interval = (−1.17, 2.90)). The means of the posteriors of the effect of FOMO and the interaction are estimated to be close to zero. Figure 8*b* shows the raw data associated with the effect of smartphone use on subsequent fatigue.

Similarly, figure 9*a* shows the posterior distributions of our model of the effect of total smartphone use on boredom. The posteriors show that if smartphone use increases by half a standard deviation, subsequent boredom is estimated to be higher by 3.94 points (95% credible interval = (2.09, 6.22)). The

[1]These sensitivity analyses are reported on the Open Science Framework page associated with this paper (https://osf.io/z9wm8/).

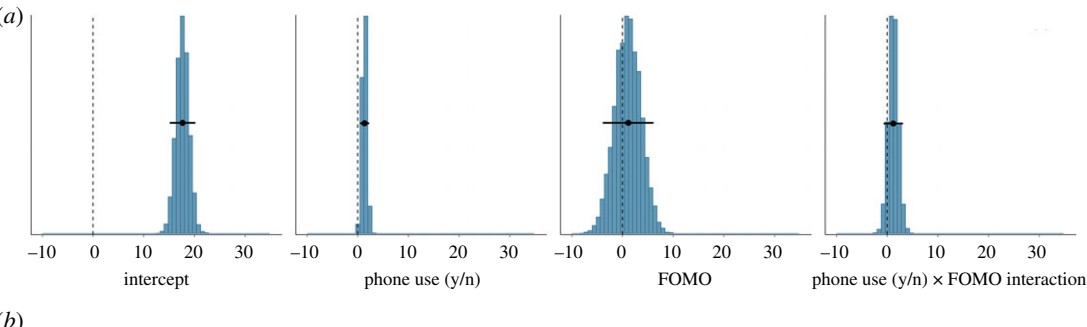

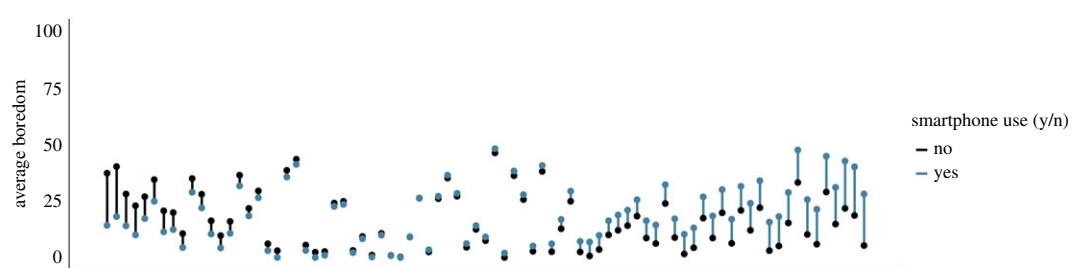

**Figure 7.** (a) The posterior distributions of the parameters for the smartphone use (y/n) model predicting subsequent boredom. The circles and the lines represent the mean of the posterior and the 95% Bayesian credible intervals, respectively. (b) The average boredom level, separately for each participant. Each participant is represented by one line. Black dots indicate participants' average boredom when they did not use their smartphone; blue dots represent participants' average boredom when they did use their smartphone. Black (blue) lines indicate that participants were more bored when they did not (did) use their smartphone. Participants are arranged by the magnitude of the relationship of smartphone use on subsequent boredom, from left (higher boredom when smartphone was not used) to right (higher boredom when smartphone was used).

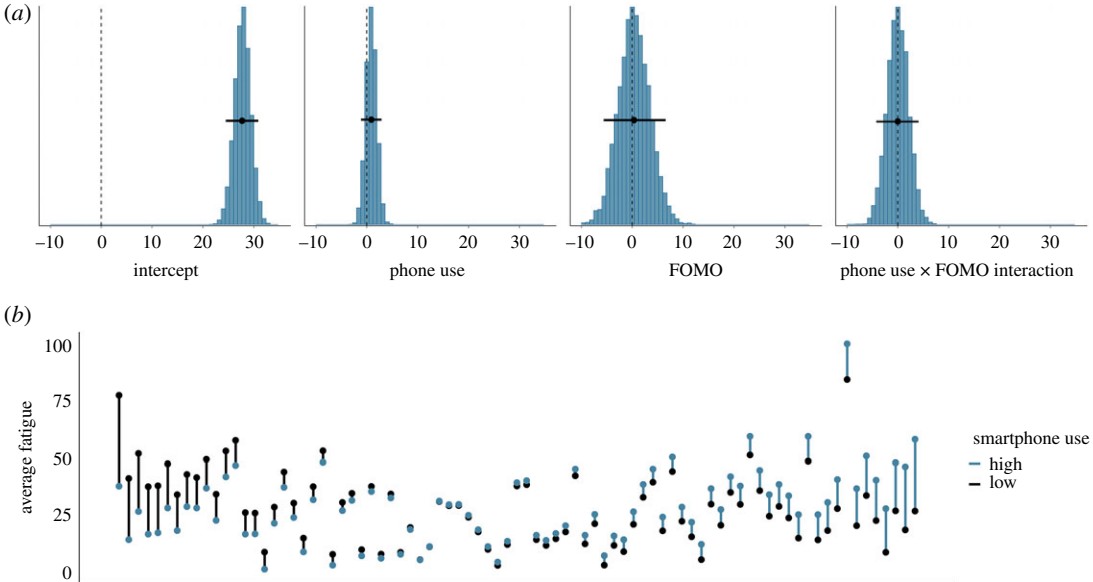

**Figure 8.** (a) The posterior distributions of the parameters for the total smartphone use model predicting subsequent fatigue. The circles and the lines represent the mean of the posterior and the 95% Bayesian credible intervals, respectively. (b) The average fatigue level, separately for each participant. Each participant is represented by one line. Black dots indicate participants' average fatigue when they used their smartphone relatively little in the previous 20 min (i.e. below their own mean level); blue dots represent participants' average fatigue when they used their smartphone relatively more in the previous 20 min (i.e. above their own mean level). Black (blue) lines indicate that participants were more fatigued when they used their smartphone less (more). Participants are arranged by the magnitude of the relationship of smartphone use on subsequent fatigue, from left (higher fatigue when smartphone use was lower) to right (higher fatigue when smartphone use was higher).

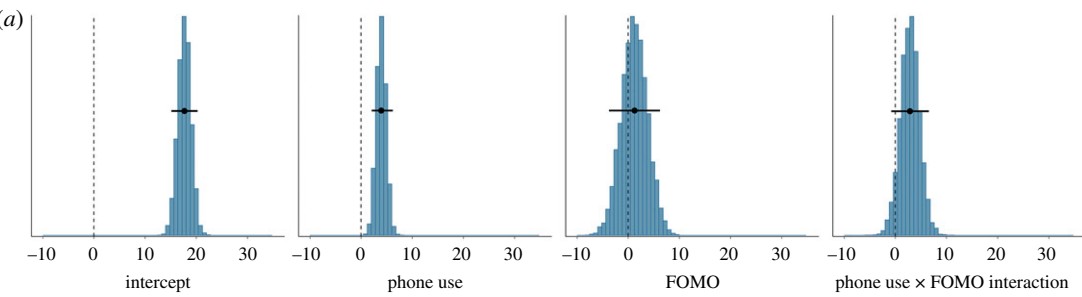

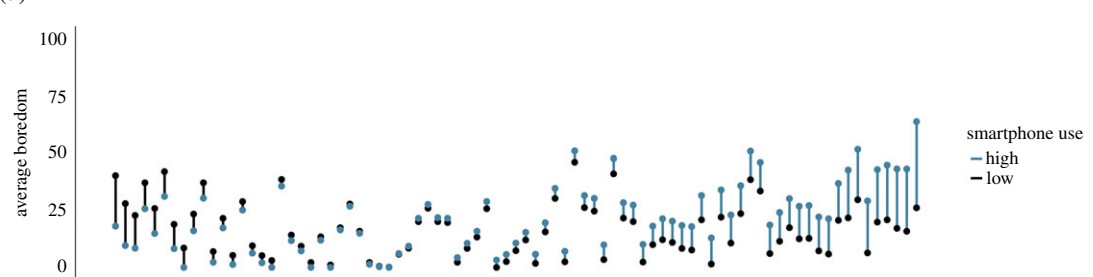

**Figure 9.** (*a*) The posterior distributions of the parameters for the total smartphone use model predicting subsequent boredom. The circles and the lines represent the mean of the posterior and the 95% Bayesian credible intervals respectively. (*b*) The average boredom level, separately for each participant. Each participant is represented by one line. Black dots indicate participants' average boredom when they used their smartphone relatively little in the previous 20 min (i.e. below their own mean level); blue dots represent participants' average boredom when they used their smartphone relatively more in the previous 20 min (i.e. above their own mean level). Black (blue) lines indicate that participants were more bored when they used their smartphone less (more). Participants are arranged by the magnitude of the relationship of smartphone use on subsequent boredom, from left (higher boredom when smartphone use was lower) to right (higher boredom when smartphone use was higher).

means of the posteriors of the effects of FOMO and the interaction are once more estimated to be close to zero. We conclude that there is some evidence for a small effect of smartphone use on the subjective experiences of fatigue and, perhaps especially, boredom. Like before, this effect is in the opposite direction to hypothesis 2. Figure 9*b* shows the raw data associated with the effect of smartphone use on subsequent boredom.

## 3. Discussion

Here, we examined the bidirectional relationship between subjective experiences that are known markers for low motivation—i.e. fatigue and boredom—and smartphone use at work, which we quantified through objective logging data. In line with hypothesis 1, findings indicate that people are more likely to switch from work to their smartphone when they are more fatigued or bored. However, we did not find evidence that people use their smartphone for longer periods of time when they are more fatigued or bored. In contrast to hypothesis 2, findings indicate that, after having used their smartphone, people experience more fatigue and boredom.

The notion that fatigue and boredom act as a 'stop emotion' [28] that trigger disengagement from the current task has been central to recent motivational models [12–15] with several laboratory studies supporting it [7,29,30]. Indeed, we found that as both fatigue and boredom increase, participants were subsequently more likely to interact with their smartphone. While the effect on the likelihood to use the smartphone looked small when interpreted visually, the model suggested that an increase in fatigue and boredom of just 10 points (on a 1–100 scale) was associated with a 30–40% increase in the likelihood of smartphone use. To put these figures in context, on average, fatigue increased by about 20 points from the morning to the afternoon. Thus, despite the fact that there are other factors that probably predict whether or not people use their smartphone (e.g. whether or not they have received a message), when people are very fatigued or bored, they seem to be substantially more likely to disengage from their work to interact with their smartphone, when compared with when they are *not* fatigued or bored. However, while fatigue was associated with the *likelihood* that people disengage from their work to interact with their smartphone, we did not find that fatigue is associated with the *duration* of smartphone

use at work. As such, our results extend previous findings that people who are more prone to experience boredom report higher smartphone use [21,22] by uncovering a similar association from one moment to the next—additionally, we show in more detail that whereas people are more likely to disengage via their smartphone when fatigued or bored, they do not spend more time on their phone.

While motivational models of fatigue assume that fatigue arises when the current task is judged to have lower value than the next-best alternative, switching from labour to leisure should allow people to regain motivation for labour, which should be accompanied by decreased experiences of fatigue (and boredom) [10,14]. One previous field study [17] found support for this idea, as the extent to which participant reported their work in the previous 90 min to be rewarding was negatively related to perceptions of fatigue. By contrast to this previous study, we found that participants reported higher fatigue and boredom after having used their smartphone. It is important to note, however, that these effects were very small, e.g. the models estimated single-digit increases in boredom and fatigue (on a 1–100 scale) when smartphone use increased by 80 s. While we cannot be certain that smartphone use *caused* the increase in boredom and fatigue, our model suggests that people do feel a slight, but sometimes notable, increase in boredom and fatigue after they have used their smartphone for several minutes in a 20 min time window.

Why could it be that people felt more fatigued and bored after having used their smartphone? We propose two possible *post hoc* explanations for this counterintuitive finding. First, if this relationship turns out to be robust and causal, it might reflect the cognitive costs that come with switching back and forth between two different tasks [31]. For example, a recent experience-sampling study suggests that multitasking at work is associated with increased feelings of mental fatigue [32]. Thus, it might not only be cognitively costly to switch between two labour tasks, but also between one labour and one leisure task. Second, it could be the case that participants' smartphone interactions were too short to boost their motivation for labour. Previous research in the laboratory [7] has shown that fatigue consistently *decreases* when people take 2 min smartphone breaks from a demanding labour task. However, in the present study, participants generally took breaks that were shorter than 2 min, which might not have the same benefit.

Our predictions rest on the assumption that the smartphone is a highly rewarding (leisure) alternative to work (labour) tasks. Especially young people, such as the participants in our sample, report smartphone use to be rewarding [33]. However, there is considerable interindividual variability in how much people enjoy to be on their phone. We tried to tap into this variability through participant's self-reported FOMO. While we did find that participants who reported high FOMO at the start of the study used their smartphone more, FOMO did not strengthen the relationships between fatigue and boredom and smartphone use. As such, there are two possibilities: (i) FOMO, even though it is related to higher smartphone use, does not reflect a higher valuation of the smartphone, or (ii) the notion that people should get fatigued faster depending on the value of the leisure alternative is false.

As we planned to test *a priori* hypotheses, in line with recent developments in the field [34], we preregistered our hypotheses, sample size and analysis plan. However, owing to several unforeseen circumstances (see Methods for more details), we decided to quit data collection before we reached our planned sample size and to follow an alternative analysis plan. As such, our results should be considered exploratory and should be interpreted with caution. Given the observational design of our study, we cannot conclude that the experiences of fatigue and boredom are causally related to smartphone use. Here, we have shown how the variables relate to one another in time in people's natural work environment. Initial experimental work [7,35] suggests that the presence of a valuable alternative to work (such as the smartphone) might increase perceptions of fatigue and boredom, and through that mechanism result in increased task disengagement. However, the same experiments suggest that people benefit from smartphone breaks more the more they enjoy these breaks. Further work is needed to integrate the results from these two lines of research. Furthermore, given the specificity of our sample, our results cannot be generalized to the whole working population. Rather, our results seem especially relevant to people working in occupations that are dominated by high mental demands. That said, assuming our findings are robust, they may point towards some important implications for employees facing high mental demands at work.

From a theoretical perspective, our study is among the first to test predictions by motivational accounts of fatigue in a real-world setting. As such, it corroborated laboratory studies that previously found fatigue to be related to task disengagement [7,29]. It also provided a first test of what happens to fatigue and boredom directly after labour-to-leisure switches. We did not find evidence for the predictions made by recent motivational models, as using the smartphone was associated with more, not less, subsequent fatigue and boredom. However, as our study is not fully conclusive (we did not

follow our preregistered analysis plan; the finding does not falsify motivational models), we suggest that more research on labour versus leisure decisions in relation to fatigue and boredom in real-life settings is necessary.

Finally, our results suggest that, rather than being a recovering *microbreak*, using one's smartphone at work might have phenomenological costs (i.e. increases in fatigue and boredom), and should thus be avoided. While we cannot establish that smartphone breaks *caused* increases in boredom and fatigue, we can cautiously conclude that smartphone breaks were associated with subsequent increases in, and not with recovery from, boredom and fatigue. As this effect was relatively small, and as it was inconsistent with the few previous studies that tested the effect of labour-to-leisure switches generally, or smartphone use more specifically, on subsequent fatigue and boredom, we caution against basing policies and interventions on this result. However, this finding is intriguing, and it warrants further exploration.

In conclusion, the present study reveals a complex interplay between fatigue and boredom and smartphone use at work. Experiencing fatigue or boredom seems to be related to an increased likelihood to interact with one's smartphone, which in turn seems to be related to an increase in experienced fatigue and boredom. So, although further research is needed, our results suggest that taking a short break with one's smartphone while at work may not have the intended positive effect on the motivation for one's work tasks.

# 4. Methods

## 4.1. Data availability and deviation from preregistration

Our data, data processing and analysis scripts, and initial preregistration are available on the Open Science Framework (OSF) (https://osf.io/z9wm8/). Two unanticipated circumstances led us to deviate from our preregistered tests. First, we did not reach our planned sample size within the timeframe that was available for this project. Second, although we planned to test our hypotheses with a mixed-level linear modelling approach, this turned out not to be viable owing to the distribution of the smartphone data (figure 1*a*). To model our data in a more appropriate manner, we opted to use a zero-inflated β model. As a consequence, we decided to additionally simplify our predictions and change the smartphone use time interval from 45 to 20 min. Given the difficulties interpreting *p*-values as analyses are changed *post hoc*, we used a Bayesian approach throughout the paper (see below). As a consequence, we no longer attempted to make binary decisions regarding our effects of interest—that is, we no longer decided whether or not an effect is statistically significant, as formalized in the null hypothesis significance testing framework. Rather, we quantified the amount of evidence for the effects of interest by conditioning the data on a *prior probability distribution*. We then present the resulting *posterior probability distribution* (figures 2–9), which communicates a range of possible effect sizes and their associated probabilities. The peak of this posterior distribution is the parameter value, which reflects the most likely effect size. The associated credible interval summarizes the width of possible effect sizes, which have a lower associated probability the farther away they move from the parameter value in either direction. In summary, we deviated from our *a priori* plan in several ways. A complete overview of our deviations as well as our reasoning can be found on the OSF project page. Thus, our results should be interpreted with caution and should be considered exploratory.

## 4.2. Participants and procedure

Initially, we set out to recruit 150 participants. We had four inclusion criteria. Participants had to (i) be employed as a PhD candidate, (ii) own an Android smartphone, (iii) self-report high job autonomy, and (iv) self-report to use their smartphone more for private matters than for work-related matters while at work. The rationale behind criterion (i) was that we wished to sample from a relatively homogeneous, young population of employees who are free to interact with their smartphone while at work. In the Netherlands, PhD candidates are employees; they usually have a 4 year, 38 h wk$^{-1}$ work contract. All participants had an office, and worked the majority of their working hours in this office. With regard to criterion (ii), participants had to own an Android smartphone as the logging application, which we used to measure smartphone use, was suitable only for the Android operating system. With criterion (iv), we did our best to ensure that the majority of smartphone interactions during our study were non-demanding leisure activities. These strict inclusion criteria, paired

with people's reluctance to share their smartphone data with researchers, made it difficult to reach our desired sample size within the timeframe that was available for this project. We decided to terminate data collection after approximately 18 months. At that point, 98 participants had completed our study. We excluded 15 participants because they either did not correctly install the application at the beginning of the study or the application was not working properly on their smartphone for other reasons, leaving us with a final sample size of 83 participants (62 female, $M_{age} = 26.78$). Participants were at varying stages in their PhD ($n_{Year\ 1} = 30$, $n_{Year\ 2} = 21$, $n_{Year\ 3} = 17$, $n_{Year\ 4} = 14$, $n_{Year\ 5} = 1$) and came from a range of faculties (e.g. $n_{social\ sciences} = 31$, $n_{medical\ sciences} = 15$, $n_{sciences} = 11$).

First, participants filled in a general questionnaire in which they reported demographics, the items measuring FOMO, as well as several questions that were meant to disguise the research questions (this took approx. 10 min). They then indicated 3 days in the following week(s) during which they planned to work from their office (meaning no work from home and no meetings, such as with supervisors or students) and downloaded the application 'App Usage—Manage/Track Usage' from the Google Play Store that tracked their smartphone usage. During the 3 days, participants received a link to a Qualtrics questionnaire (which included self-reports of fatigue and boredom) every full hour between 8.00 and 18.00 to their work email. We instructed participants to fill in these questionnaires via their computer or laptop to ensure that the questionnaires were not filled in via the smartphone. Opening and filling out each questionnaire took approximately 30 s. Participants were instructed to ignore the questionnaires they received before they arrived at work and after they left work; moreover, they were asked to fill in as many of the hourly questionnaires as possible while they were at work. These questionnaires always expired 15 min after they were sent. Additionally, at the end of the working day (18.00), participants received an end-of-day questionnaire (which addressed self-report of time they started work, time they ended work and the timing of their lunch break). This questionnaire always expired at 17.00 of the next day. During the 3 days, the application continuously logged the participants' smartphone usage. After the 3 days, participants extracted the logging data from the application and sent it to us. We then merged and anonymized the data. This procedure, as well as the entire study protocol, was approved by Radboud University's Ethics Committee Social Science (ECSW2017-1303-485), and was conducted in accordance with local guidelines. All participants gave written, informed consent to participate.

## 4.3. Measures

### 4.3.1. Fatigue and boredom

Participants rated their current level of fatigue and boredom (how fatigued do you currently feel?; how bored do you currently feel?) [36] on a 100-point Visual Analogue Scale (ranging from 'not at all' to 'extremely').

### 4.3.2. Smartphone use

The logging application data contained four variables: (i) the name of the application that is currently open (note: home screen and lock screen are considered applications by the smartphone as well), (ii) the date, (iii) the time, and (iv) the duration. Every time the open application changed, the logging application produced a new row of data. In total, this dataset contained 25 307 rows (305 application transitions per participant on average). In a first step, we preprocessed these data using PSYCHOPY [37]. Specifically, for each hourly questionnaire, we extracted the exact time when the questionnaire was submitted and we defined a pre- and post-questionnaire time interval (i.e. 20 min before and after the submission time, respectively). We then calculated the sum of the durations (in seconds) that the participant spent in applications that were not the home or lock screen in these pre- and post-questionnaire time intervals. Next, we excluded data points that had more than 2 min of overlap with (i) the reported start of the working day, (ii) the reported end of the working day, or (iii) the reported lunch break.

### 4.3.3. Fear of missing out

FOMO was assessed once in the general questionnaire. We adapted three items used in a previous study [38] measuring FOMO when using the Internet less. Our items ($\alpha = 0.75$) reflected FOMO when using the

smartphone less for private matters while at work (e.g. 'if I would use my smartphone less for private matters while at work, I would fear missing out on important things'). These items were answered on a 5-point Likert scale ranging from 1 (strongly disagree) to 5 (strongly agree).

## 4.4. Data analysis

We conducted all of our analyses in R [39]. To test the effect of fatigue and boredom on subsequent smartphone use and vice versa, we used a Bayesian (generalized) linear mixed-effects modelling approach using the *brm* function [40] (*brms* package; v. 2.10.0). In all analyses, the hourly measure was the unit of analysis. Continuous, within-subjects predictors (e.g. fatigue) were standardized within participants and then divided by 2 (so that the mean is 0 and the standard deviation is 0.5) [41]. Categorical, within-subjects predictors (e.g. whether participants used their smartphone at all) were sum-to-zero coded (−1; 1). Continuous, between-subjects predictors (e.g. FOMO) were standardized on a sample level. We aimed for 'maximal' random effects structures [42] in our models. Accordingly, we fitted three-level varying-intercept multilevel models where hourly data were nested within days of study participation, and days were nested within participants. Thus, the models contained three random intercepts for each participant, one for each day of the study. We modelled all predictors as fixed effects and random slopes varying across days and participants, except FOMO, which we assumed to be stable within participants (example R syntax: 1 + fatigue * FOMO + (1 + fatigue | participant/day)).

As smartphone use was not normally distributed and heavily zero-inflated (i.e. often participants did not interact with their smartphone at all in the post-questionnaire interval; figure 1*a*), we could not fit linear mixed-effects regressions as we had planned *a priori*. To deal with these data, we took two steps. First, we estimated the effect of fatigue and boredom on the likelihood to interact with the smartphone at all in the post-questionnaire timeframe (versus not). With this model, we test whether participants were more likely to use their smartphone at all when they were more fatigued/bored. Second, we fitted a mixed-effects model using the zero-inflated β distribution, which is appropriate for the way smartphone use was distributed in our sample [43]. In order to perform this analysis, we transformed the absolute smartphone use in seconds into the proportion of the time interval that the smartphone was used (e.g. in a 20 min interval, 120 of 1200 s equals 0.1 or 10%; for plotting and interpretation, after model fitting, we transformed the parameters once more to reflect seconds). In this mixture model, the data points in which the smartphone was not used at all are modelled separately from the data points where the smartphone was used any other amount. In this analysis, we were interested in the parameter for fatigue/boredom predicting the non-zero smartphone use. As such, with this model, we test whether participants used their smartphone more when they were more fatigued/bored in case they interacted with their smartphone at all.

As fatigue and boredom were approximately normally distributed, we could fit linear mixed-effects regression models to examine the effect of smartphone use on subsequent fatigue and boredom. To mirror the analyses for the effect of fatigue and boredom on smartphone use, we estimated the effect of whether or not the smartphone was used at all as well as the effect of total smartphone use in seconds (both in the pre-questionnaire time interval) on fatigue and boredom. Last, we tested the moderating effects of FOMO on the relationship in both directions.

For all of these analyses, we employed normally distributed weakly informative priors [41] with a mean of 0 and wide uncertainty around it, adjusted to the scale of the outcome variable, in order to be conservative. None of the results meaningfully changed when we replaced our weakly informative priors with the default brms priors. With regard to the timeframe that we use in our models, we chose a time interval of 20 min, which was meant to be a compromise between validity (the shorter the time interval, the more accurate the fatigue and boredom indicator is towards the end of the time interval) and power (the longer the time interval, the more data we have available). As sensitivity analyses, we also fitted all models in timeframes of 10 and 30 min, respectively, to make sure that the results are robust to the arbitrary timeframe that we chose. Results did not meaningfully differ as the timeframe changed from 10 to 20 to 30 min. Neither did the results differ for participants in various stages of their PhD (e.g. year 1 or year 4).

For each model, we ran four Markov chain Monte Carlo chains with 4000 samples and report the posterior distributions paired with the posterior mean and 95% credible interval. To make sure that our models converged and fit the data well, we inspected the Rhat statistic, the effective sample size, trace plots to make sure that the chains mixed, and posterior predictive checks.

Ethics. The data collection procedure and study protocol were approved by Radboud University's Ethics Committee Social Science (ECSW2017-1303-485), and was conducted in accordance with local guidelines and the Declaration of Helsinki. All participants gave written, informed consent to participate. The authors have complied with the APA ethical principles regarding research with human participants in the conduct of the research presented in this manuscript.

Data accessibility. All data that we used for our analyses as well as Python and R code for data processing, analysis, and visualization is publicly available on the Open Science Framework with identifier https://osf.io/z9wm8/.

Authors' contributions. J.D., M.v.H and E.B. designed the study concept and wrote the paper. J.D. collected and analysed the data. S.G. and M.K. revised the paper. All authors read and approved the final version of the manuscript.

Competing interests. The authors declare no competing interests.

Funding. We received no funding for this study.

Acknowledgements. We thank H. Voogd for assistance with programming the data processing scripts.

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
