## [Peer Review File · Royal Society Open Science]

Review History

RSOS-201915.R0 (Original submission)

Review form: Reviewer 1

Is the manuscript scientifically sound in its present form?

Yes

Are the interpretations and conclusions justified by the results?

No

Is the language acceptable?

Yes

Do you have any ethical concerns with this paper?

No

Have you any concerns about statistical analyses in this paper?

No

Recommendation?

Reject

Comments to the Author(s)

The authors of “Fatigue, boredom, and objectively-measured smartphone use at work” investigate if fatigue and boredom trigger smartphone use, and if smartphone use in turn serves as remedy to these two states. They tested this proposition in a diary study in which participants reported hourly boredom and fatigue, which was then correlated to smartphone use. The researchers included also ‘fear of missing out’ (FOMO) as potential moderator of these effects.

The study provided mixed support for the predictions. While both fatigue and boredom predicted greater odds to interact (vs not interact) with a smartphone, the same was not true for duration of smartphone use. Furthermore, counter to predictions, people who used (vs not used) their smartphones subsequently reported higher levels of fatigue and boredom, as opposed lower levels.

The paper deals with an interesting topic with potential practical implications. I particularly appreciate the preregistration of the study and meticulous analyses of results, alongside excellent arguments why deviation from the a priori analytical plan was warranted occasionally. However, I am not convinced that the research makes a contribution that is substantial enough, and I have qualms about the authors’ interpretation of the results. I list my main concern below.

- The theoretical contribution of the research seems limited. While I appreciate the real-life context in which fatigue and boredom are investigated, the link to smartphone use has been uncovered previously (e.g., Wolniewicz et al., 2020; Elhai et al., 2018; Ksinan et al., 2019). An examination of the proposed underlying cost-benefits process (e.g., Kurzban et al., 2013) would elevate this contribution further but, unfortunately, the researcher do not actually test these.
- The authors propose to investigate fatigue and boredom both as causal antecedent of smartphone use and as its (inverse) consequence. I agree with the authors that the resultant potential (self-regulatory?) loop is interesting to examine. The causal relationships between these variables are, in turn, a key feature of interest to the study. Yet, the study methodology does not allow for such causal inferences. It is indeed tempting to interpret temporally ordered variable measurements (smartphone use, fatigue/boredom, smartphone use) as being consistently with their causal order. Yet, this need not be the case. Without additional experimental work (or at least the inclusion of instrumental variables), assigning causality (let alone its direction) to the observed associations seems unwarranted.
- It was unclear to me why FOMO was included as putative moderator. What is the theoretical rationale for suggesting that FOMO may play this role?
- The authors should describe what fatigue, boredom, and fear of being left out are. It would also be helpful to explain to readers how the former two differ and/or overlap.
- Minor: The participant-specific data in Panels B for Figures 2-9 are difficult to interpret. Assuming that participant order is arbitrary, the authors could consider ordering these results by magnitude to ease interpretation.

Review form: Reviewer 2**Is the manuscript scientifically sound in its present form?**

Yes

Are the interpretations and conclusions justified by the results?

No

Is the language acceptable?

Yes

Do you have any ethical concerns with this paper?

No

Have you any concerns about statistical analyses in this paper?

No

Recommendation?

Accept with minor revision (please list in comments)

Comments to the Author(s)

This is a careful, interesting and thought-provoking study that breaks new ground in the real-life study of fatigue and boredom (for convenience I shall refer only to fatigue but same arguments apply to boredom). The authors chart the effect of fatigue on smartphone use, and the reverse, in a group of PhD students. In the last decade there has been a resurgence of interest in motivational theories of fatigue with fatigue serving a signal function indicating that a change from the current, fatiguing activity is required or legitimised. This contrasts with the once dominant view that fatigue was the result of the depletion of a finite resource. The authors focus on the restorative and motivational effects of switching from "labour"- working on ones thesis to "leisure"- surfing the net on a smartphone. They hypothesise, consistent with current theory, that when fatigued one will tend to switch to looking at one's smartphone and this will lead to a reduction in fatigue and return to one's labours.

This study is based on hourly measurement of fatigue and boredom on a pc delivered questionnaire while smartphone use is monitored automatically. The study was carried out over 3 working days. They had a careful, prespecified, experimental and analytic plan which required a sizeable sample (150) but recruitment and measurement difficulties enforced a major rethink in analytic method and much reduced sample of 83. They are very open about the difficulties they encountered and even with a 50% reduction in sample this is still a sizeable and demanding study with over 300 interactions with a smartphone logged and analysed for each participant. The authors present the results very fully with extensive, very helpful visualizations. The use of smartphone data is a great advance on more traditional real time studies which are often based primarily on self-report with the usual problems of common method variance and very transparent hypotheses.

They show fairly convincingly that when participants are more fatigued they are more likely to use their smartphones in the subsequent 20 minutes. Such smartphone use (a switch from labour to leisure) should lead to a return to labour and reduced fatigue. However, smartphone use in the 20 minutes prior to a fatigue measurement was associated with an increase in fatigue. The authors propose that this may be due to either the effort involved in switching between labour and leisure or briefly experienced leisure making labour appear worse and even more fatiguing. I do not find these post hoc explanations all that convincing. In addition I wonder if the study design allows one to conclude that smartphone use does indeed lead to increased fatigue. If one accepts that fatigue leads to smartphone use then it follows that participants were more fatigued than usual when they accessed their smartphone. This switch from labour to leisure may in fact have led to a reduction in fatigue but since fatigue was not assessed at the time of smartphone use (probably an impossibility) then the strength or even the presence of this effect could not be determined. When fatigue was measured up to 20 minutes later fatigue could be higher than normal but lower than it was when the smartphone was

accessed. Fatigue is highly autocorrelated over periods as long as an hour so the hourly measure will reflect fatigue over a long time period. Since the effect of smartphone use on fatigue can be examined using conventional linear MLM I suggest the authors consider allowing for this correlation by including fatigue/boredom at lag1 in their models. This might provide a more sensitive test of their hypothesis, and perhaps even allow the predicted effect to be determined.

Decision letter (RSOS-201915.R0)

Dear Dr Dora

The Editors assigned to your paper RSOS-201915 "Fatigue, boredom, and objectively-measured smartphone use at work" have now received comments from reviewers and would like you to revise the paper in accordance with the reviewer comments and any comments from the Editors. Please note this decision does not guarantee eventual acceptance.

Please submit your revised manuscript and required files (see below) no later than 21 days from today's (ie 09-Feb-2021) date. Note: the ScholarOne system will 'lock' if submission of the revision is attempted 21 or more days after the deadline. If you do not think you will be able to meet this deadline please contact the editorial office immediately.

on behalf of Essi Viding (Subject Editor)
 openscience@royalsociety.org

Associate Editor Comments to Author:
 Comments to the Author:

Your paper has now received two referee reports following peer review. Please ensure that you address these comments appropriately and re-submit your revised paper with a point-by-point response detailing any changes made.

Reviewer comments to Author:

Reviewer: 1

Comments to the Author(s)

The authors of "Fatigue, boredom, and objectively-measured smartphone use at work" investigate if fatigue and boredom trigger smartphone use, and if smartphone use in turn serves as remedy to these two states. They tested this proposition in a diary study in which participants reported hourly boredom and fatigue, which was then correlated to smartphone use. The researchers included also 'fear of missing out' (FOMO) as potential moderator of these effects.

The study provided mixed support for the predictions. While both fatigue and boredom predicted greater odds to interact (vs not interact) with a smartphone, the same was not true for duration of smartphone use. Furthermore, counter to predictions, people who used (vs not used) their smartphones subsequently reported higher levels of fatigue and boredom, as opposed lower levels.

The paper deals with an interesting topic with potential practical implications. I particularly appreciate the preregistration of the study and meticulous analyses of results, alongside excellent arguments why deviation from the a priori analytical plan was warranted occasionally. However, I am not convinced that the research makes a contribution that is substantial enough, and I have qualms about the authors' interpretation of the results. I list my main concern below.

- The theoretical contribution of the research seems limited. While I appreciate the real-life context in which fatigue and boredom are investigated, the link to smartphone use has been uncovered previously (e.g., Wolniewicz et al., 2020; Elhai et al., 2018; Ksinan et al., 2019). An examination of the proposed underlying cost-benefits process (e.g., Kurzban et al., 2013) would elevate this contribution further but, unfortunately, the researcher do not actually test these.
- The authors propose to investigate fatigue and boredom both as causal antecedent of smartphone use and as its (inverse) consequence. I agree with the authors that the resultant potential (self-regulatory?) loop is interesting to examine. The causal relationships between these variables are, in turn, a key feature of interest to the study. Yet, the study methodology does not allow for such causal inferences. It is indeed tempting to interpret temporally ordered variable measurements (smartphone use, fatigue/boredom, smartphone use) as being consistently with their causal order. Yet, this need not be the case. Without additional experimental work (or at least the inclusion of instrumental variables), assigning causality (let alone its direction) to the observed associations seems unwarranted.
- It was unclear to me why FOMO was included as putative moderator. What is the theoretical rationale for suggesting that FOMO may play this role?
- The authors should describe what fatigue, boredom, and fear of being left out are. It would also be helpful to explain to readers how the former two differ and/or overlap.

- Minor: The participant-specific data in Panels B for Figures 2-9 are difficult to interpret. Assuming that participant order is arbitrary, the authors could consider ordering these results by magnitude to ease interpretation.

Reviewer: 2

Comments to the Author(s)

This is a careful, interesting and thought-provoking study that breaks new ground in the real-life study of fatigue and boredom (for convenience I shall refer only to fatigue but same arguments apply to boredom). The authors chart the effect of fatigue on smartphone use, and the reverse, in a group of PhD students. In the last decade there has been a resurgence of interest in motivational theories of fatigue with fatigue serving a signal function indicating that a change from the current, fatiguing activity is required or legitimised. This contrasts with the once dominant view that fatigue was the result of the depletion of a finite resource. The authors focus on the restorative and motivational effects of switching from "labour"- working on ones thesis to "leisure"- surfing the net on a smartphone. They hypothesise, consistent with current theory, that when fatigued one will tend to switch to looking at one's smartphone and this will lead to a reduction in fatigue and return to one's labours.

This study is based on hourly measurement of fatigue and boredom on a pc delivered questionnaire while smartphone use is monitored automatically. The study was carried out over 3 working days. They had a careful, prespecified, experimental and analytic plan which required a sizeable sample (150) but recruitment and measurement difficulties enforced a major rethink in analytic method and much reduced sample of 83. They are very open about the difficulties they encountered and even with a 50% reduction in sample this is still a sizeable and demanding study with over 300 interactions with a smartphone logged and analysed for each participant. The authors present the results very fully with extensive, very helpful visualizations. The use of smartphone data is a great advance on more traditional real time studies which are often based primarily on self-report with the usual problems of common method variance and very transparent hypotheses.

They show fairly convincingly that when participants are more fatigued they are more likely to use their smartphones in the subsequent 20 minutes. Such smartphone use (a switch from labour to leisure) should lead to a return to labour and reduced fatigue. However, smartphone use in the 20 minutes prior to a fatigue measurement was associated with an increase in fatigue. The authors propose that this may be due to either the effort involved in switching between labour and leisure or briefly experienced leisure making labour appear worse and even more fatiguing. I do not find these post hoc explanations all that convincing. In addition I wonder if the study design allows one to conclude that smartphone use does indeed lead to increased fatigue. If one accepts that fatigue leads to smartphone use then it follows that participants were more fatigued than usual when they accessed their smartphone. This switch from labour to leisure may in fact have led to a reduction in fatigue but since fatigue was not assessed at the time of smartphone use (probably an impossibility) then the strength or even the presence of this effect could not be determined. When fatigue was measured up to 20 minutes later fatigue could be higher than normal but lower than it was when the smartphone was accessed. Fatigue is highly autocorrelated over periods as long as an hour so the hourly measure will reflect fatigue over a long time period. Since the effect of smartphone use on fatigue can be examined using conventional linear MLM I suggest the authors consider allowing for this correlation by including fatigue/boredom at lag1 in their models. This might provide a more sensitive test of their hypothesis, and perhaps even allow the predicted effect to be determined.

===PREPARING YOUR MANUSCRIPT===

===PREPARING YOUR REVISION IN SCHOLARONE===

<https://royalsociety.org/journals/authors/author-guidelines/#supplementary-material> to include a suitable title and informative caption. An example of appropriate titling and captioning may be found at [https://figshare.com/articles/Table_S2_from_Is_there_a_trade-off_between_peak_performance_and_performance_breadth_across_temperatures_for_aerobic_sc](https://figshare.com/articles/Table_S2_from_Is_there_a_trade-off_between_peak_performance_and_performance_breadth_across_temperatures_for_aerobic_scope_in_teleost_fishes_/3843624) ope_in_teleost_fishes_/3843624.

Author's Response to Decision Letter for (RSOS-201915.R0)

See Appendix A.

RSOS-201915.R1 (Revision)

Review form: Reviewer 1

Is the manuscript scientifically sound in its present form?

Yes

Are the interpretations and conclusions justified by the results?

Yes

Is the language acceptable?

Yes

Do you have any ethical concerns with this paper?

No

Have you any concerns about statistical analyses in this paper?

No

Recommendation?

Reject

Comments to the Author(s)

The authors have revised their manuscript considerably and have addressed several concerns. They added a helpful explanation of what boredom and fatigue are and how they differ, an explanation of why FOMO was expected to moderate results, clarified participant-data figures, and given a more careful description of results with respect to tentative causality. With regards to the latter, the abstract needs to be updated (currently: "our results a) provide real-life evidence that for that notion that fatigue and boredom trigger task-disengagement").

Unfortunately, my main concern, that the research currently offers a very limited theoretical contribution, remains unresolved. I appreciate that the authors highlight the methodological novelty of their approach (measuring actual smartphone behavior rather than self-reported). However, the somewhat inconclusive results do not significantly advance beyond earlier work our understanding of how smartphone use relates to fatigue and boredom:

The authors measure self-reported boredom, self-reported fatigue, and objective smartphone use. They then associate these variables. The results show positive associations between boredom and smartphone use, and fatigue and smartphone use. So far, these insights are not novel. Of course, the authors find these associations both when boredom and fatigue were measured before smartphone use, and when measured after them. As the authors acknowledge, the temporal order is not indicative of causality. In fact, an association of (a) smartphone use with subsequent boredom and subsequent fatigue may simply occur because (b) boredom and fatigue are also associated with subsequent smartphone use and instances of boredom and fatigue correlate across timepoints. Not to mention other reasons for these associations (e.g., spurious correlates).

Essentially, the study tells us that boredom and fatigue correlate with smartphone use, which is not novel. What would make this contribution theoretically substantial? One way to do so is to test the postulated, but not tested, self-regulatory process. For example, the authors could run a follow-up study in which they manipulate boredom and fatigue, followed by a measure of smartphone use. Such a study would provide insight into the tentative self-regulatory process which at present remains untested. Or the authors could test if, say, performing a task over a long time is associated with less boredom and fatigue when participants have versus do not have their smartphone with them (experimentally assigned). Unfortunately, the present contribution seems rather small.

Review form: Reviewer 2

Is the manuscript scientifically sound in its present form?

Yes

Are the interpretations and conclusions justified by the results?

Yes

Is the language acceptable?

Yes

Do you have any ethical concerns with this paper?

No

Have you any concerns about statistical analyses in this paper?

No

Recommendation?

Accept as is

Comments to the Author(s)

I am content with how you have dealt with my previous comments. The effect of smartphone use on subsequent fatigue clearly requires further study- perhaps with smartphone use triggering fatigue ratings.

Decision letter (RSOS-201915.R1)

Dear Dr Dora

On behalf of the Editors, we are pleased to inform you that your Manuscript RSOS-201915.R1 "Fatigue, boredom, and objectively-measured smartphone use at work" has been accepted for publication in Royal Society Open Science subject to minor revision in accordance with the referees' reports. Please find the referees' comments below my signature.

The Editors wanted to stress that the decision to accept with minor revision is contingent not only on responding to the reviewers' comments but also clearly discuss the limitations of the study that are outlined by the more critical of the reviewers. Given this, please do make sure you provide as full as a response as possible to this concern (both in the manuscript revision and your point-by-point response document).

Please submit your revised manuscript and required files (see below) no later than 7 days from today's (ie 05-May-2021) date. Note: the ScholarOne system will 'lock' if submission of the revision is attempted 7 or more days after the deadline. If you do not think you will be able to meet this deadline please contact the editorial office immediately.

Please note article processing charges apply to papers accepted for publication in Royal Society Open Science (<https://royalsocietypublishing.org/rsos/charges>). Charges will also apply to papers transferred to the journal from other Royal Society Publishing journals, as well as papers submitted as part of our collaboration with the Royal Society of Chemistry

(<https://royalsocietypublishing.org/rsos/chemistry>). Fee waivers are available but must be requested when you submit your revision (<https://royalsocietypublishing.org/rsos/waivers>).

on behalf of Prof Essi Viding (Subject Editor)
openscience@royalsociety.org

Reviewer comments to Author:

Reviewer: 2

Comments to the Author(s)

I am content with how you have dealt with my previous comments. The effect of smartphone use on subsequent fatigue clearly requires further study- perhaps with smartphone use triggering fatigue ratings.

Reviewer: 1

Comments to the Author(s)

The authors have revised their manuscript considerably and have addressed several concerns. They added a helpful explanation of what boredom and fatigue are and how they differ, an explanation of why FOMO was expected to moderate results, clarified participant-data figures, and given a more careful description of results with respect to tentative causality. With regards to the latter, the abstract needs to be updated (currently: "our results a) provide real-life evidence that for that notion that fatigue and boredom trigger task-disengagement").

Unfortunately, my main concern, that the research currently offers a very limited theoretical contribution, remains unresolved. I appreciate that the authors highlight the methodological novelty of their approach (measuring actual smartphone behavior rather than self-reported). However, the somewhat inconclusive results do not significantly advance beyond earlier work our understanding of how smartphone use relates to fatigue and boredom:

The authors measure self-reported boredom, self-reported fatigue, and objective smartphone use. They then associate these variables. The results show positive associations between boredom and smartphone use, and fatigue and smartphone use. So far, these insights are not novel. Of course, the authors find these associations both when boredom and fatigue were measured before smartphone use, and when measured after them. As the authors acknowledge, the temporal order is not indicative of causality. In fact, an association of (a) smartphone use with subsequent boredom and subsequent fatigue may simply occur because (b) boredom and fatigue are also associated with subsequent smartphone use and instances of boredom and fatigue correlate across timepoints. Not to mention other reasons for these associations (e.g., spurious correlates).

Essentially, the study tells us that boredom and fatigue correlate with smartphone use, which is not novel. What would make this contribution theoretically substantial? One way to do so is to test the postulated, but not tested, self-regulatory process. For example, the authors could run a follow-up study in which they manipulate boredom and fatigue, followed by a measure of smartphone use. Such a study would provide insight into the tentative self-regulatory process which at present remains untested. Or the authors could test if, say, performing a task over a long

time is associated with less boredom and fatigue when participants have versus do not have their smartphone with them (experimentally assigned). Unfortunately, the present contribution seems rather small.

===PREPARING YOUR MANUSCRIPT===

===PREPARING YOUR REVISION IN SCHOLARONE===

<https://royalsociety.org/journals/authors/author-guidelines/#supplementary-material> to include a suitable title and informative caption. An example of appropriate titling and captioning may be found at https://figshare.com/articles/Table_S2_from_Is_there_a_trade-off_between_peak_performance_and_performance_breadth_across_temperatures_for_aerobic_scops_in_teleost_fishes_/3843624.

Author's Response to Decision Letter for (RSOS-201915.R1)

See Appendix B.

Decision letter (RSOS-201915.R2)

Dear Dr Dora,

I am pleased to inform you that your manuscript entitled "Fatigue, boredom, and objectively-measured smartphone use at work" is now accepted for publication in Royal Society Open Science.

on behalf Essi Viding (Subject Editor)
openscience@royalsociety.org

RSOS-201915

Title: Fatigue, boredom, and objectively-measured smartphone use at work

Corresponding author: Jonas Dora

Point-by-point response to Reviewer #1

Reviewer: 1

Comments to the Author(s)

The authors of “Fatigue, boredom, and objectively-measured smartphone use at work” investigate if fatigue and boredom trigger smartphone use, and if smartphone use in turn serves as remedy to these two states. They tested this proposition in a diary study in which participants reported hourly boredom and fatigue, which was then correlated to smartphone use. The researchers included also ‘fear of missing out’ (FOMO) as potential moderator of these effects.

The study provided mixed support for the predictions. While both fatigue and boredom predicted greater odds to interact (vs not interact) with a smartphone, the same was not true for duration of smartphone use. Furthermore, counter to predictions, people who used (vs not used) their smartphones subsequently reported higher levels of fatigue and boredom, as opposed lower levels.

The paper deals with an interesting topic with potential practical implications. I particularly appreciate the preregistration of the study and meticulous analyses of results, alongside excellent arguments why deviation from the a priori analytical plan was warranted occasionally. However, I am not convinced that the research makes a contribution that is substantial enough, and I have qualms about the authors’ interpretation of the results. I list my main concern below.

[1] The theoretical contribution of the research seems limited. While I appreciate the real-life context in which fatigue and boredom are investigated, the link to smartphone use has been uncovered previously (e.g., Wolniewicz et al., 2020; Elhai et al., 2018; Ksinan et al.,

2019). An examination of the proposed underlying cost-benefits process (e.g., Kurzban et al., 2013) would elevate this contribution further but, unfortunately, the researcher do not actually test these.

We thank Reviewer #1 for this comment and for pointing our attention to these previous papers. We believe that our paper provides a contribution over these previous papers for two important reasons. First, in our paper, we quantified smartphone use by using software that objectively recorded when and how long participants interacted with their smartphones, whereas the studies mentioned by the reviewer all assessed smartphone use by using self-reports. Previous research has convincingly shown that self-reports of smartphone use do not correlate well with actual behavior (Scharrow, 2016, Communication Methods and Measures). Hence, our work goes beyond previous research by providing a more rigorous test of the temporal relationship between fatigue/boredom and smartphone use. We understand, however, that this novel contribution may not have been sufficiently explicit in the previous version of our manuscript. So, we have added the following sentences to our introduction to highlight this contribution of our research more clearly, see pp. 4 – 5:

To test our hypotheses, we conducted an experience-sampling study in which participants (PhD candidates who owned an Android smartphone, reported high job autonomy, and reported to use their smartphone more for private matters than for work-related matters during working hours) rated their current level of fatigue and boredom, every full hour while they were at work, for three working days. At the same time, an application on participants' smartphone continuously monitored their smartphone use. **The use of such a monitoring application is a deviation from most previous research on the antecedents and consequences of smartphone use, which has generally relied on self-reports of smartphone use (e.g., Elhai, Levine, Dvorak, & Hall, 2016; Wolniewicz, Rozgonjuk, & Elhai, 2020). We nevertheless chose to use a monitoring application, because previous research (Scharrow, 2016) has shown that self-reported smartphone use does not correlate highly with actual smartphone use. Thus, by linking self-report data on fatigue and boredom with objective smartphone use data, we were able to model the temporal relationship of these affective states with objective smartphone use, and vice versa.**

Furthermore, we agree with Reviewer #1 that it is interesting to examine the proposed underlying cost-benefits process (e.g., Kurzban et al., 2013). We did indeed present analyses in our paper that directly speak to this underlying cost-benefit computation. In the revised version, we have clarified this contribution as well, as we explain below (see Reviewer #1, Comment [3], where we discuss FOMO's role as a moderator).

[2] The authors propose to investigate fatigue and boredom both as causal antecedent of smartphone use and as its (inverse) consequence. I agree with the authors that the resultant potential (self-regulatory?) loop is interesting to examine. The causal relationships between these variables are, in turn, a key feature of interest to the study. Yet, the study methodology does not allow for such causal inferences. It is indeed tempting to interpret temporally ordered variable measurements (smartphone use, fatigue/boredom, smartphone use) as being consistently with their causal order. Yet, this need not be the case. Without additional experimental work (or at least the inclusion of instrumental variables), assigning causality (let alone its direction) to the observed associations seems unwarranted.

We thank Reviewer #1 for raising this important point. We fully agree that we cannot establish causality with the design of our study; we can only show the temporal relationship between fatigue/boredom and smartphone use (and vice versa). In line with this limitation, in the previous version of the paper, we attempted to avoid causal language throughout the paper and focused on the temporal relationship (e.g., 'In line with this idea, we hypothesize that (more) fatigue and boredom is related to (more) subsequent smartphone use at work (hypothesis 1).'; 'The posteriors show that as fatigue increases by half a standard deviation (~ 11 points), participants are estimated to be 1.36 times more likely to interact with their smartphone in the following 20 minutes (95% credible interval = [1.02, 1.82).'; 'In line with hypothesis 1, findings indicate that people are more likely to switch from work to their smartphone when they are more fatigued or bored.').

However, to address Reviewer #1's comment, we have critically re-read our entire paper and agree with Reviewer #1 that the discussion of our results regarding hypothesis 2 (effect of smartphone use on subsequent fatigue/boredom) could be interpreted as implying causation. We

agree that we cannot conclude that using the smartphone more leads to higher feelings of fatigue and boredom. We have adapted our language in the discussion to clarify this, see pp. 18 & 20:

While motivational models of fatigue assume that fatigue arises when the current task is judged to have lower value than the next-best alternative, switching from labor to leisure should allow people to regain motivation for labor, which should be accompanied by decreased experiences of fatigue (and boredom)^{10,14}. One previous field study¹⁶ found support for this idea, as the extent to which participant reported their work in the previous 90 minutes to be rewarding was negatively related to perceptions of fatigue. By contrast to this previous study, we found that participants reported higher fatigue and boredom after having used their smartphone. It is important to note, however, that these effects were very small; e.g., the models estimated single-digit increases in boredom and fatigue (on a 1–100 scale) when smartphone use increased by 80 seconds. **While we cannot be certain that smartphone use *caused* the increase in boredom and fatigue, our model suggests that people do feel a slight, but sometimes noticeable, increase in boredom and fatigue after they have used their smartphone for several minutes in a 20-minute time window.**

[...]

Finally, our results suggest that, rather than being a recovering *microbreak*, using one's smartphone at work might have phenomenological costs (i.e., increases in fatigue and boredom), and should thus be avoided. **While we cannot establish that smartphone breaks *caused* increases in boredom and fatigue, we can cautiously conclude that smartphone breaks were associated with subsequent increases in, and not with recovery from, boredom and fatigue.** As this effect was relatively small, and as it was inconsistent with the few previous studies that tested the effect of labor-to-leisure switches generally, or smartphone use more specifically, on subsequent fatigue and boredom, we caution against basing policies and interventions on this result. However, this finding is intriguing, and it warrants further exploration.

To avoid further potential causal interpretation of our findings, we have added the following sentences to our discussion of limitations, see p. 20:

As we planned to test a priori hypotheses, in line with recent developments in the field²⁸,

we preregistered our hypotheses, sample size, and analysis plan. However, due to several unforeseen circumstances (see Method for more details), we decided to quit data collection before we reached our planned sample size and to follow an alternative analysis plan. As such, our results should be considered exploratory and should be interpreted with caution. **Given the observational design of our study, we cannot conclude that the experiences of fatigue and boredom are causally related to smartphone use. Here, we have shown how the variables relate to one another in time. Further experimental work is needed to test whether these temporal relationships have a causal underlying mechanism (for initial experimental work, see Dora et al., 2019; Rom, Katzir, Diel, & Hofmann, 2019).** Furthermore, given the specificity of our sample, our results cannot be generalized to the whole working population. Rather, our results seem especially relevant to people working in occupations that are dominated by high mental demands. That said, assuming our findings are robust, they may point towards some important implications for employees facing high mental demands at work.

Finally, to completely avoid any causal interpretation of our results, we replaced 'predicts with 'is related to' in the formulation of hypothesis 1.

[3] It was unclear to me why FOMO was included as putative moderator. What is the theoretical rationale for suggesting that FOMO may play this role?

We thank Reviewer #1 for pointing out to us that the theoretical reason to include FOMO in the study was not yet clear. Modern theoretical accounts of fatigue/boredom (e.g., the opportunity cost model, Kurzban et al., 2013) suggest that the experiences of fatigue/boredom during labor do not only depend on characteristics of the labor task, but also on the value of the leisure alternative (e.g., the smartphone). For that reason, our hypothesized relationships should be expected to be stronger for participants who value their smartphone more. FOMO is an individual difference in the degree to which people value online interactions, which should reflect the value of the smartphone. We have clarified this reasoning in the introduction section, see p. 5:

Our predictions rest on the assumption that the smartphone is a highly valued leisure alternative to labor (Kurzban et al., 2013). More specifically, this opportunity cost model predicts that the relationship between the subjective experiences of fatigue and boredom depend on the value of the leisure alternative to work (e.g., the smartphone). Thus, our predicted effects should be stronger for participants who value their smartphone (interactions) more. Research indicates that people differ in their desire to stay continuously connected to their friends and family via the internet²⁰. This individual difference, labeled *fear of missing out* (FOMO), is thought to reflect the degree to which people value to stay connected to others through digital technology²⁰. Hence, higher FOMO should be associated with a higher value of the current leisure alternative. For that reason, we additionally tested whether individual differences in fear of missing out (FOMO) strengthen the relationships of smartphone use with fatigue and boredom in both directions.

As we write in the discussion (p. 19), not finding the interaction with FOMO in our study could imply either of two things:

- *people reporting higher FOMO do not actually value their smartphone more, or;*
- *valuing the smartphone more has no effect on the experiences of fatigue and boredom.*

Further experimental work will be needed to dissociate these two remaining explanations.

[4] The authors should describe what fatigue, boredom, and fear of being left out are. It would also be helpful to explain to readers how the former two differ and/or overlap.

We thank Reviewer #1 for pointing out to us that we should clarify the concepts we are studying. We have rewritten the relevant parts in our introduction section and have added the similarities and differences between fatigue and boredom according to motivational theories of these experiences, see pp. 3 – 4 & p. 5:

Our hypotheses are grounded in recent motivational models of fatigue and boredom. We assume that to continue working vs. to use one's smartphone represents a goal conflict—specifically, a conflict between *labor goals* and *leisure goals*^{10,11}. In the context of cognitive

work, a labor task is any activity that is productive but mentally demanding (e.g., grading a thesis); a leisure task is any activity that is unproductive and mentally undemanding (e.g., answering a friend's text message). According to motivational models of fatigue and boredom, the shared function of these experiences is to resolve such goal conflicts^{10,12-15}. Specifically, fatigue and boredom should arise when the current (labor) task is judged to have lower value than some alternative (leisure) task. In other words, conscious feelings of fatigue and boredom are thought to reflect a discrepancy between what is currently being done and what should be done instead. Both fatigue and boredom are defined as aversive subjective experiences (Eastwood, Frischen, Fenske, & Smilek, 2012; Hockey, 2011). The difference between fatigue and boredom is assumed to depend on the amount of stimulation currently provided by the (labor) task. When people invest a lot of cognitive resources into the current task while this task is judged to be less valuable than some alternative, people should experience fatigue¹⁴; when the current task provides insufficient cognitive stimulation and the current task is judged to be less valuable than some alternative, people should experience boredom¹⁵.

[...]

Our predictions rest on the assumption that the smartphone is a highly valued leisure alternative to labor (Kurzban et al., 2013). More specifically, this opportunity cost model predicts that the relationship between the subjective experiences of fatigue and boredom depend on the value of the leisure alternative to work (e.g., the smartphone). Thus, our predicted effects should be stronger for participants who value their smartphone (interactions) more. Research indicates that people differ in their desire to stay continuously connected to their friends and family via the internet²⁰. This individual difference, labeled *fear of missing out* (FOMO), is thought to reflect the degree to which people value to stay connected to others through digital technology²⁰. Hence, higher FOMO should be associated with a higher value of the current leisure alternative. For that reason, we additionally tested whether individual differences in fear of missing out (FOMO) strengthen the relationships of smartphone use with fatigue and boredom in both directions.

[5] Minor: The participant-specific data in Panels B for Figures 2-9 are difficult to

interpret. Assuming that participant order is arbitrary, the authors could consider ordering these results by magnitude to ease interpretation.

We thank Reviewer #1 for this comment. We realized that the previous visualization was not as clear as we had hoped, which is why we changed these plots. Now, each participant is represented by one line which reflects the difference in the DV as the IV is high or low. In line with Reviewer #1's suggestion, we ordered the results by magnitude, see e.g. Figure 2:

Figure 2. Panel A shows the exponentiated posterior distributions of the parameters (reflecting the odds ratios) for the fatigue model predicting subsequent likelihood to use the smartphone. The circles and the lines represent the mean of the posterior and the 95% Bayesian credible intervals respectively. **Panel B** shows the probability of smartphone use in the 20-minute time frame, separately for each participant. Each participant is represented by one line. Black dots indicate participants' probability of smartphone use when they are relatively low in fatigue (i.e., below their own mean level); blue dots represent participants' probability of smartphone use when they are relatively high in fatigue (i.e., above their own mean level). Black (blue) lines indicate that participants were more likely to use the smartphone when fatigue was low (high). Participants are arranged by the magnitude of the relationship of fatigue on subsequent likelihood to use the smartphone, from left (higher probability to use smartphone when fatigue is lower) to right (higher probability to use smartphone when fatigue is higher).

Point-by-point response to Reviewer #2

Reviewer: 2

Comments to the Author(s)

This is a careful, interesting and thought-provoking study that breaks new ground in the real-life study of fatigue and boredom (for convenience I shall refer only to fatigue but some arguments apply to boredom). The authors chart the effect of fatigue on smartphone use, and the reverse, in a group of PhD students. In the last decade there has been a resurgence of interest in motivational theories of fatigue with fatigue serving a signal function indicating that a change from the current, fatiguing activity is required or legitimized. This contrasts with the once dominant view that fatigue was the result of the depletion of a finite resource. The authors focus on the restorative and motivational effects of switching from “labour”- working on one’s thesis to “leisure”- surfing the net on a smartphone. They hypothesise, consistent with current theory, that when fatigued one will tend to switch to looking at one’s smartphone and this will lead to a reduction in fatigue and return to one’s labours.

This study is based on hourly measurement of fatigue and boredom on a pc delivered questionnaire while smartphone use is monitored automatically. The study was carried out over 3 working days. They had a careful, prespecified, experimental and analytic plan which required a sizeable sample (150) but recruitment and measurement difficulties enforced a major rethink in analytic method and much reduced sample of 83. They are very open about the difficulties they encountered and even with a 50% reduction in sample this is still a sizeable and demanding study with over 300 interactions with a smartphone logged and analyzed for each participant. The authors present the results very fully with extensive, very helpful visualizations. The use of smartphone data is a great advance on more traditional real time studies which are often based primarily on self-report with the usual problems of common method variance and very transparent hypotheses.

We thank Reviewer #2 for these nice words.

[1] They show fairly convincingly that when participants are more fatigued they are more likely to use their smartphones in the subsequent 20 minutes. Such smartphone use (a

switch from labour to leisure) should lead to a return to labour and reduced fatigue. However, smartphone use in the 20 minutes prior to a fatigue measurement was associated with an increase in fatigue. The authors propose that this may be due to either the effort involved in switching between labour and leisure or briefly experienced leisure making labour appear worse and even more fatiguing. I do not find these post hoc explanations all that convincing.

We thank Reviewer #2 for this comment and for challenging our post-hoc explanations. We have further substantiated our explanations using relevant and recent literature. With regard to possible switch costs, we now refer to Kudesia et al., 2020, who show in an experience-sampling study that people experience more mental fatigue at work when they multitask. We have also simplified our second explanation, by contrasting our finding to previous laboratory experiments that showed that people did become less fatigued when they took smartphone breaks of 2 minutes. While we thus think these explanations are plausible in principle, we agree with Reviewer #2 that they are also speculative, which is what we now communicate in this paragraph more clearly, see p. 19:

Why could it be that people felt more fatigued and bored after having used their smartphone? We propose two possible post-hoc explanations for this counterintuitive finding. First, if this relationship turns out to be robust and causal, it might reflect the cognitive costs that come with switching back and forth between two different tasks²⁶. For example, a recent experience-sampling study suggests that multitasking at work is associated with increased feelings of mental fatigue (Kudesia, Pandey, & Reina, 2020). Thus, it might not only be cognitively costly to switch between two labor tasks, but also between one labor and one leisure task. Second, it could be the case that participants' smartphone interactions were too short to boost their motivation for labor. Previous research in the laboratory (Dora et al., 2019) has shown that fatigue consistently *decreases* when people take two-minute smartphone breaks from a demanding labor task. However, in the present study, participants generally took breaks that were shorter than two minutes, which might not have the same benefit.

[2] In addition, I wonder if the study design allows one to conclude that smartphone use does indeed lead to increased fatigue. If one accepts that fatigue leads to smartphone use

then it follows that participants were more fatigued than usual when they accessed their smartphone. This switch from labour to leisure may in fact have led to a reduction in fatigue but since fatigue was not assessed at the time of smartphone use (probably an impossibility) then the strength or even the presence of this effect could not be determined. When fatigue was measured up to 20 minutes later fatigue could be higher than normal but lower than it was when the smartphone was accessed. Fatigue is highly autocorrelated over periods as long as an hour so the hourly measure will reflect fatigue over a long time period. Since the effect of smartphone use on fatigue can be examined using conventional linear MLM I suggest the authors consider allowing for this correlation by including fatigue/boredom at lag1 in their models. This might provide a more sensitive test of their hypothesis, and perhaps even allow the predicted effect to be determined.

We thank Reviewer #2 for proposing this sensitivity analysis. We had done and reported something similar previously, in that we controlled for time of day in these analyses, since we observed an increase in mental fatigue over the working day. This sensitivity analysis did not reveal significant changes in our parameter estimates.

We understand the rationale for controlling for lagged fatigue. However, we chose against reporting this analysis, as it resulted in a substantial loss of observations (about 20% of observations need to be excluded, because [a] there is no fatigue measure prior to the start of the working day, meaning we cannot predict fatigue in the first hour of work and [b] we lose observations in cases participants did not fill out consecutive questionnaires). That said, we did re-run our models in line with Reviewer #2's recommendation. These analyses revealed that the effect of smartphone use on fatigue is slightly smaller, but did not change in directionality. The effect of boredom was unchanged (if anything, it got slightly bigger). Regarding these analyses, it is important to keep in mind that they were run on 80% of our sample.

Taken these two sensitivity analyses together, and given our already careful interpretation of the small effect of smartphone use on fatigue, we conclude that these sensitivity analyses do not meaningfully change our conclusions. We have uploaded a document to our osf page (<https://osf.io/z9wm8/>) reporting the parameter estimates and 95% CIs of the lagged models for you and other readers to inspect. In the manuscript, we now point the reader to this alternative robustness analysis in footnote 1. We hope you and Reviewer #2 agree with this solution.

RSOS-201915

Title: Fatigue, boredom, and objectively-measured smartphone use at work

Corresponding author: Jonas Dora

Point-by-point response to Reviewer #1

Reviewer: 1

Comments to the Author(s)

The authors have revised their manuscript considerably and have addressed several concerns. They added a helpful explanation of what boredom and fatigue are and how they differ, an explanation of why FOMO was expected to moderate results, clarified participant-data figures, and given a more careful description of results with respect to tentative causality. With regards to the latter, the abstract needs to be updated (currently: “our results a) provide real-life evidence that for that notion that fatigue and boredom trigger task-disengagement”).

We are happy to hear that Reviewer #1 agrees with most of the changes we made in our previous revision. We have updated our abstract to reflect the changed wording of the discussion of Hypothesis 1. Specifically, we now write that our results provide real-life evidence for the notion that fatigue and boredom “are temporally associated with” task disengagement.

Unfortunately, my main concern, that the research currently offers a very limited theoretical contribution, remains unresolved. I appreciate that the authors highlight the methodological novelty of their approach (measuring actual smartphone behavior rather than self-reported). However, the somewhat inconclusive results do not significantly advance beyond earlier work our understanding of how smartphone use relates to fatigue and boredom:

The authors measure self-reported boredom, self-reported fatigue, and objective smartphone use. They then associate these variables. The results show positive associations between boredom and smartphone use, and fatigue and smartphone use. So far, these insights are not novel (**point 1**). Of course, the authors find these associations both when boredom and fatigue were measured before smartphone use, and when measured after them. As the authors acknowledge, the

temporal order is not indicative of causality. In fact, an association of (a) smartphone use with subsequent boredom and subsequent fatigue may simply occur because (b) boredom and fatigue are also associated with subsequent smartphone use and instances of boredom and fatigue correlate across timepoints **(point 2)**. Not to mention other reasons for these associations (e.g., spurious correlates).

Essentially, the study tells us that boredom and fatigue correlate with smartphone use, which is not novel. What would make this contribution theoretically substantial? One way to do so is to test the postulated, but not tested, self-regulatory process. For example, the authors could run a follow-up study in which they manipulate boredom and fatigue, followed by a measure of smartphone use. Such a study would provide insight into the tentative self-regulatory process which at present remains untested. Or the authors could test if, say, performing a task over a long time is associated with less boredom and fatigue when participants have versus do not have their smartphone with them (experimentally assigned). **(point 3)** Unfortunately, the present contribution seems rather small.

To summarize, in the more critical part of their review, Reviewer #1 raises three points.

1) Previous research already showed an association between fatigue/boredom and smartphone use. So, the present study is not novel.

2) We find that smartphone use predicts subsequent fatigue/boredom. However, this association may be spurious, because:

- a. fatigue and boredom are associated with subsequent smartphone use;
- b. fatigue and boredom correlate with subsequent fatigue and boredom;

So, the present study does not provide evidence of causality.

3) If the present study would have been complemented by experimental work, conclusions would have been stronger.

We will now address these three points directly, and explain how Reviewer #1's feedback has led to changes in the manuscript.

1) Not novel

In their initial review, Reviewer #1 pointed out three papers (Elhai et al., 2018; Ksinan et al., 2019; Wolniewicz et al., 2020). These papers each report a cross-sectional survey study, in which ‘boredom proneness’ (a between-participant measure of one’s general proneness to experience boredom) correlated with a general self-report of smartphone use (from ‘never’ to ‘very often’). The studies do not address fatigue. Reviewer #1 argued that these studies have already uncovered the link between fatigue/boredom and smartphone use. Thus, Reviewer #1 concluded that our findings are not novel.

In our view, our study goes beyond this prior work in important ways:

- Rather than relying on cross-sectional data, our intensive ecological momentary assessment can describe how fatigue/boredom and smartphone use are temporally associated from one moment to the next (within-persons, rather than between-persons). Our approach also allows us to differentiate effects on the likelihood and duration of use.
- We measure smartphone use objectively rather than relying on people’s self-report. This is important, because people are highly inaccurate when self-reporting their smartphone use, and self-reports of smartphone use do not meaningfully correlate with objectively logged smartphone use (e.g., Burnell et al., 2021).

For these reasons, we politely disagree with the conclusion that our study and results are not novel, as our design affords a much more valid analysis of the fatigue/boredom – smartphone use association compared to prior work on the topic. That said, we understand Reviewer #1’s point and we wish to do it justice. We thus made the following changes in the manuscript:

First, we summarized the findings of these three previous studies and further clarified the differences with our study in the introduction section:

The use of momentary measures of fatigue and boredom as well as a monitoring application to quantify smartphone allows us to improve on previous research indicating correlations between subjective experiences and smartphone use (e.g., Elhai, Levine, Dvorak, & Hall, 2016; Wolniewicz, Rozgonjuk, & Elhai, 2020). These studies, which suggested that participants who generally experience more boredom also use their smartphone more, relied on cross-sectional assessments of subjective experiences and self-reported smartphone use. Previous research (Burnell, 2021; Scharrow, 2016) has shown that self-reported

smartphone use does not correlate meaningfully with actual smartphone use. Thus, by linking state-level self-report data on fatigue and boredom with momentary objective smartphone use data, we were able to model the moment-to-moment temporal relationship of these affective states with objective smartphone use within participants, and vice versa.

Moreover, in the discussion, we explicitly mention how our results relate to previous studies on the fatigue/boredom – smartphone use association:

[...] However, while fatigue was associated with the *likelihood* that people disengage from their work to interact with their smartphone, we did not find that fatigue is associated with the *duration* of smartphone use at work. As such, our results extend previous findings that people who are more prone to experience boredom report higher smartphone use (Elhai et al., 2018, Wolniewicz et al., 2020) by uncovering a similar association from one moment to the next – additionally, we show in more detail that whereas people are more likely to disengage via their smartphone when fatigued or bored, they do not spend more time on their phone.

2) Spurious, not causal

Regarding the alternative explanations Reviewer #1 provides for the counter-intuitive finding of smartphone use on subsequent fatigue and boredom, we have shown through our sensitivity analyses (as part of our previous revision) that they are unlikely to explain our results. Most notably, the fact that our results do not change when we control for fatigue/boredom in the previous hour makes the explanation that fatigue and boredom have auto-correlation implausible. We agree, however, that this point was not yet sufficiently clear in the previous version of the manuscript. So, we now clarify this contribution of our sensitivity analysis in the results section:

Figure 6A shows the posterior distributions of our model of the effect of whether or not the smartphone was used on fatigue. The posteriors show that if the smartphone was used in the 20 minutes before the hourly questionnaire, fatigue is estimated to be higher by 0.89 points (95% credible interval = [-0.14, 1.91]). The effects of FOMO on fatigue and the interaction between whether or not the smartphone was used and FOMO are estimated to be close to zero. These results (as well as the results from the subsequent three models) did not change meaningfully when we controlled for time of day or fatigue at the previous hour, showing that the effect is not due to general increases in fatigue and boredom over the

course of the day or in the previous hour¹. Figure 6B shows the raw data associated with the effect of whether or not the smartphone was used on subsequent fatigue.

3) Experimental work needed

We agree with Reviewer #1 that experimental studies are needed to clarify whether the temporal association found here might be causal. Indeed, we did several laboratory experiments that are along the lines of what Reviewer #1 suggests. Specifically, in four experiments (in press at *Journal of Experimental Psychology: General*), we let participants choose to either work on a paid cognitively demanding task or take an unpaid break. To examine the opportunity cost model, we manipulated the reward value of this break, providing participants the chance to interact with their own smartphone during the break or with other, less rewarding, activities. Participants then repeatedly chose to either work or take a break while we continuously measured their fatigue level. We found and replicated that participants got more fatigued while working and recovered more while taking a break the more they enjoyed the alternative to work (such as the smartphone).

So, while we fully agree that experiments would help complement our study here, we also stand by our choice to report these experiments in a separate paper. To illustrate, our recent report in *JEP:G* has 10,000 words, and tells quite a nuanced story already. We feel that these two reports are best read separately, especially since they cater to different audiences. For example, compared to our report in *JEP:G*, the present report is more relevant to audiences that care about intensive measures of behavior in real-life settings. To address this issue, we now discuss our results here in light of our experimental work in more detail in the discussion section:

[...] Given the observational design of our study, we cannot conclude that the experiences of fatigue and boredom are causally related to smartphone use. Here, we have shown how the variables relate to one another in time in people's natural work environment. Initial experimental work (Dora et al., 2021; Rom et al., 2019) suggests that the presence of a valuable alternative to work (such as the smartphone) might increase perceptions of fatigue and boredom, and through that mechanism result in increased task disengagement. However, the same experiments suggest that people benefit from

¹ These sensitivity analyses are reported on the OSF page associated with this paper (<https://osf.io/z9wm8/>).

smartphone breaks more the more they enjoy these breaks. Further work is needed to integrate the results from these two lines of research. [...]